# Unified Off-Policy Learning to Rank: a Reinforcement Learning Perspective

**Zeyu Zhang**[1]    **Yi Su**[2*]   **Hui Yuan**[3]    **Yiran Wu**[4]    **Rishab Balasubramanian**[5]
**Qingyun Wu**[4]    **Huazheng Wang**[5*]    **Mengdi Wang**[3]

[1]University of Science and Technology of China    [2]Google Deepmind    [3]Princeton University
[4]Penn State University    [5]Oregon State University

zgkd2019zzy@mail.ustc.edu.cn    yisumtv@google.com
{huiyuan, mengdiw}@princeton.edu    {ykw5399, qingyun.wu}@psu.edu
{balasuri, huazheng.wang}@oregonstate.edu

## Abstract

Off-policy Learning to Rank (LTR) aims to optimize a ranker from data collected by a deployed logging policy. However, existing off-policy learning to rank methods often make strong assumptions about how users generate the click data, i.e., the click model, and hence need to tailor their methods specifically under different click models. In this paper, we unified the ranking process under general stochastic click models as a Markov Decision Process (MDP), and the optimal ranking could be learned with offline reinforcement learning (RL) directly. Building upon this, we leverage offline RL techniques for off-policy LTR and propose the Click Model-Agnostic Unified Off-policy Learning to Rank (CUOLR) method, which could be easily applied to a wide range of click models. Through a dedicated formulation of the MDP, we show that offline RL algorithms can adapt to various click models without complex debiasing techniques and prior knowledge of the model. Results on various large-scale datasets demonstrate that CUOLR consistently outperforms the state-of-the-art off-policy learning to rank algorithms while maintaining consistency and robustness under different click models.

## 1    Introduction

Learning to Rank (LTR) is a core problem in Information Retrieval (IR) with wide applications such as web search and recommender systems [33]. Traditional LTR methods require high-quality annotated relevance judgments for model training, which is expensive, time-consuming, and may not align with actual user preferences [41]. As a result, learning to rank with implicit user feedback, such as logged click data, has received a huge amount of attention in both academia and industry [23, 26, 53].

Despite its low cost, learning to rank directly from implicit user feedback could suffer from the intrinsic noise and bias in user interactions, e.g., position bias, where an item displayed at a higher position receives a higher click-through rate (CTR) than its relevance [12]. To mitigate the bias in the click data, off-policy learning to rank methods have been proposed under different bias assumptions such as position bias [53, 2], selection bias [52] and trust bias [36, 9]. A major branch of off-policy learning to rank called counterfactual learning to rank achieves unbiasedness by re-weighting the samples using the inverse propensity scoring (IPS) method [46, 2, 27]. To estimate the propensity from logged click data, existing works require explicit assumptions about how users examine the rank list and generate the click data, i.e., click models [12, 10]. For example, position-based model

---

*Correspondence to: Yi Su and Huazheng Wang

(PBM) [24] assumes the probability of examining a result only depends on the position; while cascade model (CASCADE) [12] assumes each click depends on the previous click, and dependent click model (DCM) [18] considers both. Different debiasing methods have been proposed to cator to specific click models, including PBM [4, 53, 26], CASCADE; and DCM [48]. However, prior knowledge of the click model is usually unknown and the correct click model needs to be identified from user behavior data before applying an off-policy algorithm, which is challenging in complex real-world environments. Besides, many popular and powerful click models have not been studied in counterfactual learning to rank such as the click chain model (CCM) [17] and the user browsing model (UBM) [13]. It requires a significant amount of work to study debiasing methods for every popular click model.

To overcome these issues, we propose to study a unified approach of off-policy learning to rank adaptable to general click models. Our key insight is that the user's examination and click behavior summarized by click models has a Markov structure; thus off-policy LTR under general click models can be formulated as a *Markov Decision Process (MDP)*. Specifically, the learning to rank problem now can be viewed as an episodic RL problem [45, 1], where each time step corresponds to a ranking position, each action selects a document for the position, and the state captures the user's examination tendency. This formulation allows us to view off-policy LTR from the perspective of *offline reinforcement learning* [31], where we can leverage off-the-shelf offline RL algorithms [29, 19, 16] to optimize the ranking list. Importantly, our formulation bridges the area of off-policy learning to rank and offline RL, allowing for the integration of ideas and solutions from offline RL to enhance the solution of the off-policy LTR problem.

Inspired by the formulation, we propose the Click Model-Agnostic Unified Off-policy Learning to Rank (CUOLR) method. We first construct each logged query and ranking data as an episode of reinforcement learning following the MDP formulation. Our dedicated structure for state representation learning can efficiently capture the dependency information for examination and click generation, e.g. ranking position in PBM and previous documents in CM and DCM. The algorithm jointly learns state representation and optimizes the policy, where any off-the-shelf offline RL algorithm can be applied as a plug-in solver. Specifically, we adapt the popular CQL algorithm [29] as an instantiation, which applies the conservative (pessimism) principle to Q function estimate. We evaluate our algorithm on real-world learning to rank datasets [38, 7] under various click models. Compared with off-policy LTR methods that are dedicated to specific click models, our click model-agnostic method consistently outperforms the best-performing baselines in all click models.

The contributions of this paper are summarized as follows:

- We formulate the off-policy LTR with biased feedback under general click model as a Markov Decision Process, and bridge the area of off-policy learning to rank and offline reinforcement learning.

- We propose CUORL, a Click model-agnostic Unified Off-policy LTR method that could utilize any offline RL algorithm as a plug-in solver, and we instantiate it using CQL.

- We conduct extensive empirical experiments to validate the effectiveness of our algorithm using real-world LTR datasets under different click models.

## 2 Related Work

**Off-policy Learning to Rank.** Off-policy Learning to Rank aims to optimize the ranking function from logged click data [23]. The majority of the works aim to mitigate the bias in logged click data, known as counterfactual learning to rank or unbiased learning to rank. The debiasing methods mainly follow inverse propensity scoring strategy [26, 49, 4, 51, 48, 3], while there are also recent works applying doubly robust estimator to reduce variance [40, 28, 37]. Cief et al. [11] proposed pessimistic off-policy optimization for learning to rank that also mitigates bias but not in an unbiased way. All these methods rely on prior knowledge of the click model [12, 10], while our algorithm is agnostic to general click models.

**Offline Reinforcement Learning.** Offline RL algorithms [29, 14, 44, 21, 16, 15] learn policy from large logged datasets where the distributional shift between the logging policy and the learned policy imposes a major challenge. In this setting, different algorithms are proposed, from value-based ones

(e.g. [35, 34]) to policy-based ones (e.g. [43, 42]). Among the vast literature on offline reinforcement learning, the principle of pessimism/conservatism [29, 5, 22] is an important line and has inspired many algorithms from empirical and theoretical perspective [60, 39, 32, 56, 61, 55, 58]. While all the aforementioned methods can be plugged into our algorithm, we choose the classic CQL algorithm [29] with a conservative Q function on top of soft actor-critic algorithm [19].

**Reinforcement Learning to Rank.** Wei et al. [54] first model ranking problem as an MDP, where the state is the candidate document set at current rank and the action is the selected document. [57, 62] have been studied under similar MDP formulation. However, [54, 57] requires relevance labels as feedback and cannot mitigate bias in click data; [62] is an online learning algorithm that learns from user interactions instead of logged data. Compared to these studies, we characterize the MDP formulation from a different perspective, i.e., capture bias in the click model, and propose the offline RL algorithm with logged click feedback.

# 3 Reinforcement Learning to Rank: A Unified Formulation

As the majority of existing works in unbiased learning to rank focused on inferring documents' relevance from the click models, these methods are tied to specific click models adopted. In this section, we formulate learning to rank with general click feedback as a Markov decision process, offering a unified and comprehensive modeling of possibly complicated user behavior. This formulation unifies LTR problems associated with different click models, under which the underlying click model is translated into the environment setup of MDP such as state transitions and rewards. It opens up the possibility to employ a rich class of reinforcement learning algorithms for solving LTR, which we will give greater details in the next section.

## 3.1 Preliminary

**Click model.** A key challenge of off-policy LTR lies in learning the document's attractiveness/relevance from the implicit feedback that is biased by the user's examination behavior. To address this challenge, a range of click models have been proposed to accommodate user's various click behavior [10]. In this study, we focus on a general family of click models [63, 30], which is marked by two characteristics: (1) Most of the mainstream click models have a "two-step" flavor that breaks user's click behavior towards some document down into the user's examination and the document's relevance. For each document $d$, the user first decides whether to examine it in the ranking list, based on the specific behavior. Mathematically the user behavior is modeled as the examination probability, which generally depends on the ranking list $\mathcal{R}$ and the position of the document $k$, denote as $\chi(\mathcal{R}, k)$. Once the document is examined, the user will choose whether to click it, based on the attractiveness $\alpha(d)$ [2]. (2) Any documents under the $k$-th position do not have an effect on $\chi(\mathcal{R}, k)$.

**Definition 1** (Click Model). *For any rank list $\mathcal{R}$ and position $k$, the attractiveness and examination probability are independent.*

$$P(C_k = 1 \mid \mathcal{R}, k) = \chi(\mathcal{R}, k)\alpha(\mathcal{R}(k)) \tag{1}$$

*where $C_k$ is the click indicator at rank $k$, and $\chi(\mathcal{R}, k)$ is the examination probability of position $k$ in the rank list $\mathcal{R}$. For each document $d$, the attractiveness $\alpha(d)$ only depends on the document itself. And the attractiveness is mutually independent.*

We show that classic click models such as PBM, CASCADE, DCM, and CCM are instances of Definition 1, with details listed in Appendix C.

## 3.2 Learning to Rank as Markov Decision Process

In Reinforcement Learning (RL), the interactions between the agent and the environment are often described as an *finite-horizon discounted Markov Decision Process $M = (S, A, T, r, \gamma, H)$*. The goal of the RL problem is to find a policy $\pi$ that maximizes the value function, i.e. the discounted

---

[2]We simplify the notation and assume $d$ captures (query, document) pair information on given query.

cumulative reward

$$\mathbb{E}\left[\sum_{t=0}^{H}\gamma^t r(s_t, a_t) \mid \pi, s_0 = s\right]. \tag{2}$$

In what follows, we formulate each of the $(S, A, T, r, \gamma, H)$ components in the ranking scenario. Our formulation essentially differs from the existing MDP formulation of ranking [54], where the state at position $k$ is defined as remaining documents that are yet to rank following the $k-1$ ranked ones on the top, instead Wei et al. [54] has a limited capturing of user's click behavior as a result of being ignorant of the ordering within the top $k-1$ documents. From here on, we use $k \in [K]$ to denote the $k$-th position top down on a list of total length $K$. It serves as the counterpart of the time step $t$ in (2) for the ranking setting.

**State** $\mathcal{S}$    For each position $k \in [K]$, state $s_k$ should include and represent the current status of the ranking that the agent is faced with. Thus, we define the state at rank $k$ as:

$$s_k = [(d_1, d_2, \ldots, d_{k-1}), k], \tag{3}$$

which is a concatenation of the established sub-ranking list up to $k$, denoted by $(d_1, d_2, \ldots, d_{k-1})$, and the position $k$. Here $d_i$ refers to the document presented at rank $i$ with $s_0$ is initialized as $= [(), 0]$. Together with the action $a_k$ as to select document $d_k$ presenting at rank $k$, defining $s_k$ as (3) fully captures the user's click behavior $C_k$ at this point. Recall (1) that $P(C_k = 1 \mid \mathcal{R}, k) = \chi(\mathcal{R}, k)\alpha(\mathcal{R}(k))$, where $\chi(\mathcal{R}, k)$ is determined by $(d_1, d_2, \ldots, d_{k-1})$. To better capture the rich information in the observed state, we discuss how to attain the effective state embedding from the raw representation in Section 4.

**Action** $\mathcal{A}$    Action $a_k$ is naturally defined as the document to present at rank $k$ given state $s_k$. In our experiments, each action is represented by a feature vector associated with the query. It is worth mentioning that the available action set $\mathcal{A}_k$ at each $k$ is related to state $s_k$ as well as the specific query, unlike the common case where the action space is fixed. There are two main differences here compared with the fixed action space: (1). the available actions vary under different queries; and (2). once an action is chosen, it is removed from the candidate set $\mathcal{A}_k$.

**Transition** $\mathcal{T}(s'|s, a)$    Transition maps a state-action pair to the probability distribution over possible next states. Given our formulation of state and action, the next state $s_{k+1}$ is deterministic given $s_k = [(d_1, d_2, \cdots, d_{k-1}), k]$ and $a_k$. Formally, $\mathcal{T}(s_{k+1}|s_k, a_k) = 1$ if and only if $s_{k+1} = [(d_1, d_2, \cdots, d_{k-1}, a_k), k+1]$. Note that with this transition function, we can show that our model is a rigorous MDP, where the distribution of next state only based on the current state and the action.

**Reward** $r(s, a)$    Aligned with the goal of LTR to maximize total clicks, we adopt the binary click as a reward at each position, i.e. $r(s_k, a_k) = C_k$. It is easily checked this is a well-defined reward from (1) that the distribution of $r$ is fully determined by $s$ and $a$, i.e., $\mathbb{E}[r(s_k, a_k)] = \chi(s_k)\alpha(a_k)$.

Putting the components together we have formally built up our MDP formulation of LTR, which we name as "**MDP for Ranking**" and denote by $\mathcal{MR}(\mathcal{S}, \mathcal{A}, \mathcal{T}, r, \gamma, H)$ with components defined as above. The rich RL literature has the convention to solve the optimal policy $\pi^*$ that maximizes the cumulative reward, which in our proposed MDP translates to

$$\pi^* = \underset{\pi}{\operatorname{argmax}} \mathbb{E}\left(\sum_{k=1}^{K}\gamma^{k-1} r(s_k, \pi(\cdot \mid s_k))\right), \tag{4}$$

where the expectation is taken over the stochasticity in both environments $\mathcal{MR}$ and policy $\pi$. Before leveraging any RL algorithms for ranking, it is necessary to validate good policies of $\mathcal{MR}(\mathcal{S}, \mathcal{A}, \mathcal{T}, r, \gamma, K)$ yields good ranking lists. In the following subsection, we give a rigorous theorem for this validation.

### 3.3    Optimizing Rank List by Optimizing Policy

**Constructing a rank list given policy** $\pi$.    With the definition of MDP, we can construct a rank list with any given policy sequentially. At each position $k$, the state is constructed based on previous document features (manually or by sequential models). Then a document (action) is chosen by the policy $\pi$ and placed at position $k$, where the next state is determined accordingly. Repeat this process at each position until all the documents are set or the list reaches its end ($K = 10$ for example).

**Definition 2** (Policy Induced Ranking.). *Given a policy $\pi(\cdot \mid s)$ of $\mathcal{MR}(\mathcal{S}, \mathcal{A}, \mathcal{T}, r, \gamma)$, construct the induced rank list $\mathcal{R}^\pi$ as*

$$\mathcal{R}^\pi(k) \leftarrow a_k \sim \pi(\cdot \mid s_k).$$

To investigate whether the optimal rank list can be captured by optimal policy $\pi^*$, we start with defining the optimality of rank list.

**Definition 3** (Optimality). *A rank list $\mathcal{R}$ is optimal if and only if all documents are sorted in descending order in terms of their attractiveness, i.e.*

$$\alpha(\mathcal{R}(1)) \geq \ldots \geq \alpha(\mathcal{R}(K))$$

*and $\{\mathcal{R}(1), \cdots, \mathcal{R}(K)\}$ are the $K$ most attractive documents among all, where $K$ is the length of the list and it is also called the top-K ranking.*

**Assumption 3.1** (Optimality of optimal ranking). *Let $V_\mathcal{R}(s_1) = \mathbb{E}[\sum_{k=1}^K \gamma^{k-1} r(s_k, \mathcal{R}(k))]$ be the value of rank list $\mathcal{R}$, and let $\mathcal{R}^\star$ be the optimal rank list by Definition 3. Then $\max_\mathcal{R} V_\mathcal{R}(s_1) = V_{\mathcal{R}^\star}(s_1)$.*

Assumption 3.1 is adopted from Assumption 2 in [30], which suggests optimal rank list sorted by decreasing attractiveness of documents leads to optimal rewards, which will be covered by optimal policy learned from our MDP formulation. This is a mild assumption as classic click models such as PBM and cascade model all satisfy the assumption [63].

# 4   Unified Off-policy Learning to Rank

The formulation of off-policy learning-to-rank (LTR), when viewed from the perspective of offline reinforcement learning, presents an opportunity to leverage off-the-shelf RL algorithms to address off-policy LTR problems. In this section, we introduce a novel and unified off-policy LTR algorithm that is agnostic to the underlying click model used to generate the offline training data. Our algorithm is composed of three key components: (1) episodes constructed from the logged ranking data; (2). state representation learning; and (3). policy optimization via offline RL algorithms. In the following, we provide a detailed exposition of each component.

**Episodes Construction.**   Given any logged ranking dataset, the first step is to construct a series of episodes from the ranking data, making it compatible with the off-the-shelf RL algorithms. Specifically, our original ranking data $\{(q_i, \mathcal{R}_i, \mathbf{c}_i(q_i, \mathcal{R}_i))\}_{i=1}^n$ is composed of $n$ tuples with contextual query feature $q_i$, a ranked list $\mathcal{R}_i$ with $K$ documents and a corresponding click vector $\mathbf{c}_i(q_i, \mathcal{R}_i) \in \{0, 1\}^K$. From the perspective of RL, this tuple can be conceptualized as one episode, following the established MDP formulation of the ranking process. In particular, for each tuple $(q_i, \mathcal{R}_i, \mathbf{c}_i(q_i, \mathcal{R}_i))$, we transform it to a length $K$ episode $\tau_i := \{(s_k^i, a_k^i, r_k^i)\}_{k=1}^K$ with

$$s_k^i := \phi(\mathcal{R}_i[: k], k), \quad a_k^i := \mathcal{R}_i[k], \quad r_k^i := \mathbf{c}_i(q_i, \mathcal{R}_i)[k] \quad \text{for } \forall k \in [K]$$

Here we use $\mathcal{R}_i[: k]$ to denote the concatenation of the document feature vectors before position $k$ and $\mathcal{R}_i[k]$ represents the document feature at position $k$, similarly for $\mathbf{c}_i(q_i, \mathcal{R}_i)[k]$. In particular, the episode $\tau_i$ is constructed by going through the ranked list from top to the bottom. Each state $s_k^i$ contains all the document information before position $k$ and the position information $k$, represented as a function of $\mathcal{R}_i[: k]$ and $k$ with $\phi$ being a learned embedding function that we will discuss shortly. The action at step $k$ is the document placed at position $k$, i.e., $\mathcal{R}_i[k]$, with the reward at current timestep is the binary click for the corrpsonding action at position $k$, i.e., $\mathbf{c}_i(q_i, \mathcal{R}_i)[k]$. Given this, we have constructed an offline RL dataset with $n$ episodes with length $K$ each.

**State Representation Learning.**   In RL, state representation learning offers an efficient means of managing large, raw observation spaces, enhancing the generalization performance when encountering previously unseen states. In our particular setting, the observation space consists of raw document features up to position $k$. As we will show in Section 5.2, utilizing raw observations as the state representation in the RL algorithm can lead to highly sub-optimal performance, due to both unscalability with respect to $k$ and limitations in the representation power. Rather than incorporating additional auxiliary tasks to explicitly learn state representations, we propose to jointly learn the state

representations together with the policy optimization algorithm, which aims to automatically learn state representation $\phi$ that will benefit the downstream policy optimization task. For example, DQN uses multiple layers of nonlinear functions to encode the map perceptual inputs to a state embedding that could be linearly transformed into the value function. To this end, we introduce the *implicit state representation learning component* our off-policy learning to rank algorithm, which is composed of the following key components:

*Positional Encoding*: To effectively inject the position information in $s_k^i$, we utilize the *positional encoding* technique [50, 59], ensuring the model make use of the position information when generating clicks. Specifically, positional encoding represents each position $k$ in the ranked list by the sinusoidal function with different frequencies such that

$$PE(k)_{2i} = \sin\left(\frac{k}{10000^{2i/d_{\text{model}}}}\right), \quad PE(k)_{2i+1} = \cos\left(\frac{k}{10000^{2i/d_{\text{model}}}}\right),$$

Here $PE(k) \in \mathbb{R}^d$ with $d$ being the dimension of document feature and $2i$ and $2i+1$ being the even and the odd entries of $PE(k)$ with $2i, 2i+1 \in [d]$.

*Multi-head Self-attention*: The other challenge in our case is to find a specific architecture for the state representation learning that tailored for the learning to rank task. As the ranked list of document features is inherently sequential, we leverage the multi-head self-attention mechanism to learn the state embedding $\phi$. Specifically, the state $s_k^i$ is defined as:

$$s_k^i := \phi(\mathcal{R}_i[:k], k) = \text{Concat}(head_1, \ldots, head_I)W^O$$

where $head_i = \text{Attention}(s_k \cdot W_i^Q, s_k \cdot W_i^K, s_k \cdot W_i^V)$, with $W_i^Q, W_i^K, W_i^V \in \psi_i$ are learnable parameters for the $i^{th}$ head and $W^O$ is the learnable parameter of the output layer after concatenating the results of all the $I$ heads; At each position $k$, we concatenate the document features $\mathcal{R}_i[:k]$ along with the position embedding for position $k$, passing them as the input for the multi-head self-attention.

**Joint Representation Learning and Policy Optimization.** Given the constructed offline dataset and a dedicated structure for learning efficient state representations, we now demonstrate how we leverage any off-the-shelf RL algorithm as the plug-in solver for finding the optimal policy. We use the popular offline RL algorithm: CQL, as an instantiation, which learns a conservative Q function and utilizes the classical soft actor-critic algorithm on top of it. Specifically, it optimizes the following lower bound of the critic (Equation 5):

$$\hat{\theta} \leftarrow \underset{\theta}{\arg\min} \, \alpha \mathbb{E}_{s \sim \mathcal{D}}\left[\log \sum_a \exp(Q_\theta(s, a)) - \mathbb{E}_{a \sim \pi_\beta(a|s)}[Q_\theta(s, a)]\right] + \frac{1}{2}\mathbb{E}_{s,a,s' \sim \mathcal{D}}\left[\left(Q_\theta - \hat{\mathcal{B}}^\pi \hat{Q}_{\theta'}\right)^2\right]$$
$$(5)$$

where $\hat{\mathcal{B}}^\pi Q = r + \gamma P^\pi Q$ is the estimated *Bellman operator*, $\pi_\beta$ is the logging policy. Here we use $\theta'$ to emphasize that the parameters in the target Q network is different from policy Q network. The conservative minimizes the expected Q value for out-of-distribution (state, action) pairs and prevents the Q-function's over-estimation issue. Built upon SAC, the algorithm improves the policy $\pi_\xi$ (i.e., the actor) based on the gradient of the estimated Q function, with entropy regularization. Compared with the original CQL algorithm, we also add the state representation learning component, and we jointly optimize the state embedding, the critic and actor parameters, with a detailed algorithm included in Algorithm 1. Once we have obtained the learned optimal policy $\pi_\xi^*$, we extract the optimal ranking policy following Definition 2.

## 5 Experiments

We empirically evaluate the performance of our proposed method CUOLR on several public datasets, compared with the state-of-the-art off-policy learning-to-rank methods. Specifically, we aim to assess the robustness of our method across different click models and showcase the effectiveness of our unified framework. Beyond this, we perform the ablation study to examine the efficacy of our proposed state representation learning component. [3].

---

[3]Codes: `https://github.com/ZeyuZhang1901/Unified-Off-Policy-LTR-Neurips2023`

---

**Algorithm 1** Click Model-Agnostic Unified Off-policy Learning to Rank (with CQL)

---

1: **Inputs:** logged ranking data $\{(q_i, \mathcal{R}_i, \mathbf{c}_i(q_i, \mathcal{R}_i))\}_{i=1}^{n}$, length of the ranking $K$, batch size $B$, train iteration $T$.
2: **Initialize:** policy $\pi_\xi$ and Q function $Q_\theta$, embedding model $\phi_\psi(\cdot, \cdot)$
3: **for** $t \in [T]$ **do**:
4:     Randomly sample a batch of queries $\mathcal{Q}$ with size $B$.
5:     Construct offline RL episodes $\mathcal{T} = \left\{ \{(s_k^i, a_k^i, r_k^i)\}_{k=1}^{K} \right\}_{i:q_i \in \mathcal{Q}}$.

$$s_k^i = \phi_\psi(\mathcal{R}_i[:k], k), \quad a_k^i = \mathcal{R}_i[k], \quad r_k^i = \mathbf{c}_i(q_i, \mathcal{R}_i)[k] \quad \text{for } \forall k \in [K]$$

6:     Train the Q-net (and embedding model) with loss defined in equation (5)

$$\theta \leftarrow \theta - \eta_Q \nabla_\theta \text{Loss}(\theta, \mathcal{T})$$
$$\psi \leftarrow \psi - \eta_\phi \nabla_\psi \text{Loss}(\theta, \mathcal{T})$$

7:     Improve policy $\pi_\xi$ (and embedding model) with SAC-style entropy regularization.

$$\xi \leftarrow \xi + \eta_\pi \mathbb{E}_{s \sim \mathcal{T}, a \sim \pi_\xi(\cdot|s)} [Q_\theta(s, a) - \log \pi_\xi(a \mid s)]$$
$$\psi \leftarrow \psi + \eta_\phi \mathbb{E}_{s \sim \mathcal{T}, a \sim \pi_\xi(\cdot|s)} [Q_\theta(s, a) - \log \pi_\xi(a \mid s)]$$

8: **end for**
9: **Output:** learned ranking policy $\pi_\xi^*$, Q function $Q_\theta^*$, embedding model $\phi_\psi^*(\cdot, \cdot)$
10: Recover the optimal ranking from the learned policy $\pi_\xi^*$ using Definition 2.

---

## 5.1 Setup

**Datasets.** We conduct semi-synthetic experiments on two traditional learning-to-rank benchmark datasets: MSLR-WEB10K and Yahoo! LETOR (set 1). Specifically, we sample click data from these real-world datasets, which not only increases the external validity of the experiments but also provides the flexibility to explore the robustness of our method over different click models. For both datasets, it consists of features representing query and document pairs with manually judged relevance labels ranging from 0 (irrelevant) to 4 (perfectly relevant). We provide the statistics of all datasets in Appendix A. Both datasets come with the train-val-test split. The train data is used for generating logging policy and simulating clicks, with the validation data used for hyperparameter selection. And the final performance of the learned ranking policy is evaluated in the test data.

**Click Data Generation.** We follow Joachims et al. [25] to generate partial-information click data from the full-information relevance labels. Specifically, we first train a Ranking SVM [26] using $1\%$ of the training data as our logging policy $\pi_\beta$ to present the initial ranked list of items. For each query $q_i$, we get a ranking $\mathcal{R}_i$ and simulate the clicks based on various click models we use. As discussed in Section 3.1, there are two components for the click generation, examination probability and attractiveness of the document for the query. All click models differ in their assumptions on the examination probability. For PBM, we adopt the examination probability $\boldsymbol{\rho} = \{\rho_k\}_{k=1}^{K}$ estimated by Joachims et al. [25] through eye-tracking experiments:

$$\chi(\mathcal{R}^q, k) = \chi(k) = \rho_k^\eta$$

where $\eta \in [0, +\infty]$ is a hyper-parameter that controls the severity of presentation biases and in our experiment, we set $\eta = 1.0$ as default. For CASCADE, the examination probabilities are only dependent on the attractions of each previous document. For DCM, the $\lambda$s are also simulated by the same parameters as PBM examination probability. We use the same attraction models for all click models, as defined following:

$$\alpha(d) = \epsilon + (1 - \epsilon)\frac{2^{r(d)} - 1}{2^{r_{max}} - 1}$$

where $r(d) \in [0, 4]$ is the relevance label for document $d$ and $r_{max} = 4$ is the maximum relevance label. We also use $\epsilon$ to model click noise so that irrelevant documents have a non-zero probability to be treated as attractive and being clicked.

Table 1: Performance comparison with different click models on Yahoo! LETOR set1 and MSLR-WEB10K. "*" and "**" indicate statistically significant improvement (p-value < 0.05 and p-value < 0.01 respectively) over the best baseline for each metric respectively.

| CLICK MODEL | ALG | Yahoo! LETOR | | | | MSLR-WEB10K | | | |
| | | ERR@K | | NDCG@K | | ERR@K | | NDCG@K | |
| | | K=5 | K=10 | K=5 | K=10 | K=5 | K=10 | K=5 | K=10 |
|---|---|---|---|---|---|---|---|---|---|
| PBM | DLA | 0.439 | 0.455 | 0.691 | 0.742 | 0.256 | 0.278 | 0.356 | 0.384 |
| | CM-IPW | 0.440 | 0.456 | 0.692 | 0.743 | 0.255 | 0.277 | 0.354 | 0.383 |
| | IPW | 0.446 | 0.462 | **0.700** | 0.748 | 0.281 | 0.301 | 0.367 | 0.390 |
| | CUOLR(CQL) | **0.458**\*\* | **0.473**\*\* | **0.700** | **0.753**\* | **0.281** | **0.303** | **0.380**\*\* | **0.406**\*\* |
| | CUOLR(SAC) | **0.459**\*\* | **0.478**\*\* | **0.700** | **0.753**\*\* | 0.279 | **0.301** | **0.379**\*\* | **0.404**\*\* |
| CASCADE | DLA | 0.441 | 0.457 | 0.690 | 0.741 | 0.265 | 0.286 | 0.365 | 0.392 |
| | CM-IPW | 0.444 | 0.460 | **0.696** | 0.745 | **0.283** | **0.304** | **0.378** | 0.403 |
| | IPW | 0.442 | 0.457 | 0.690 | 0.740 | 0.257 | 0.279 | 0.358 | 0.387 |
| | CUOLR(CQL) | **0.459**\*\* | **0.479**\*\* | **0.696** | **0.748** | 0.280 | 0.301 | **0.378** | **0.404** |
| | CUOLR(SAC) | **0.461**\* | **0.478**\*\* | **0.696** | **0.748** | 0.279 | 0.301 | **0.379** | **0.405** |
| DCM | DLA | 0.444 | 0.459 | 0.696 | 0.745 | 0.279 | 0.299 | 0.378 | 0.404 |
| | CM-IPW | 0.447 | 0.463 | **0.704** | 0.752 | **0.284** | **0.304** | 0.379 | 0.402 |
| | IPW | 0.444 | 0.459 | 0.697 | 0.746 | 0.278 | 0.299 | 0.363 | 0.387 |
| | CUOLR(CQL) | **0.461**\*\* | **0.474**\*\* | 0.699 | **0.753** | 0.278 | 0.300 | **0.380** | **0.405** |
| | CUOLR(SAC) | **0.461**\*\* | **0.475**\*\* | 0.703 | **0.755** | 0.278 | 0.300 | 0.378 | 0.403 |
| | LOGGING | 0.385 | 0.403 | 0.630 | 0.693 | 0.206 | 0.230 | 0.304 | 0.338 |
| | ORACLE | **0.462** | **0.476** | **0.739** | **0.781** | **0.328** | **0.347** | **0.432** | **0.453** |

**Baselines and Hyperparameters.** We compare our CUOLR method with the following baselines: (1). *Dual Learning Algorithm (DLA)* [4] which jointly learns an unbiased ranker and an unbiased propensity model; (2). *Inverse Propensity Weighting (IPW)* Algorithm [52, 26] which first learns the propensities by result randomization, and then utilizes the learned probabilities to correct for position bias; and (3). *Cascade Model-based IPW (CM-IPW)* [48] which designs a propensity estimation procedure where previous clicks are incorporated in the estimation of the propensity. It is worth mentioning that (1) and (2) are designed for PBM and (3) is tailored for cascade-based models. Besides, we train a LambdaMart [6] model with true relevance labels as the upper bound for the ranking model, *ORACLE* for short. The performance of the logging policy (*LOGGING*) is also reported as the lower bound of the ranking model.

For baselines, we use a 2-layer MLP with width 256 and ReLU activation according to their original paper and codebase [4, 51, 47]. For the embedding model in our method, we use multi-head attention with 8 heads. And for actors and critics in CQL and SAC algorithms, we utilize a 2-layer MLP with width 256 and ReLU activation. The conservative parameter $\alpha$ (marked red in Equation (5)) in CQL is set to 0.1. We use Adam for all methods with a tuned learning rate using the validation set. More details are provided in Appendix A.

**Metrics.** We evaluate all the methods using the full-information test set. We use the *normalized Discounted Cumulative Gain (nDCG)* [20] and the *Expected Reciprocal Rank (ERR)* [8] as evaluation metrics and report the results at position 5 and 10 to demonstrate the performance of models at different positions.

## 5.2 Results

**How does CUOLR perform across different click models, compared to the baselines?** To validate the effectiveness of our CUOLR method, we conducted performance evaluations across a range of click-based models, including PBM, CASCADE, DCM, and CCM. We compared our approach with state-of-the-art baselines specifically designed for these click models, namely DLA, IPW, and CM-IPW. Due to space limitation, we only show results of PBM, CASCADE, and DCM in Table 1, with the full table shown in Appendix B. For the position-based model, IPW demonstrates the

Table 2: Performance of CUOLR algorithm with different state embeddings. The experiments are conducted on Yahoo! LETOR set1 and MSLR-WEB10K, with PBM and CASCADE as click models.

| CLICK MODEL | STATE EMBED | Yahoo! LETOR | | | | MSLR-WEB10K | | | |
| | | ERR@K | | NDCG@K | | ERR@K | | NDCG@K | |
| | | K=5 | K=10 | K=5 | K=10 | K=5 | K=10 | K=5 | K=10 |
|---|---|---|---|---|---|---|---|---|---|
| PBM | POS | 0.439 | 0.455 | 0.691 | 0.742 | **0.282** | **0.304** | **0.382** | **0.407** |
| | PREDOC | 0.435 | 0.451 | 0.682 | 0.734 | 0.274 | 0.295 | 0.374 | 0.401 |
| | POS + PREDOC | 0.440 | 0.456 | 0.685 | 0.734 | 0.277 | 0.298 | 0.375 | 0.400 |
| | ATTENTION | **0.459** | **0.478** | **0.700** | **0.753** | 0.281 | 0.303 | 0.380 | 0.406 |
| CASCADE | POS | 0.426 | 0.442 | 0.668 | 0.724 | 0.260 | 0.283 | 0.360 | 0.391 |
| | PREDOC | 0.456 | 0.472 | **0.703** | **0.754** | 0.272 | 0.293 | 0.373 | 0.400 |
| | POS+PREDOC | 0.453 | 0.470 | 0.686 | 0.740 | 0.275 | 0.296 | 0.373 | 0.396 |
| | ATTENTION | **0.461** | **0.478** | 0.696 | 0.748 | **0.280** | **0.301** | **0.378** | **0.404** |
| | ORACLE | **0.462** | **0.486** | **0.739** | **0.781** | **0.328** | **0.347** | **0.432** | **0.453** |

best performance among the baselines, which is expected as it is tailored for position-based methods. Similarly, CM-IPW yielded the best performance for the cascade-based methods, which aligns with its incorporation of previous document information in the propensity estimation. Remarkably, across all click models, our method, whether combined with SAC or CQL, consistently achieves the best performance in most cases. This validates the effectiveness of our unified framework and the robustness of our CUOLR algorithm. Furthermore, it is noteworthy that our method demonstrated consistent performance across different RL algorithms, verifying its resilience and adaptability to various underlying RL solvers.

**How effective is the state representation learning component in CUOLR?** In this experiment, we examine different approaches for state representation learning and study how it affects the overall performance of our proposed method. We compare with the following state embeddings: (1). position-only embedding (POS), which only utilizes the position information using positional encoding; (2). previous-document-based embedding (PREDOC), which takes a simple average of all the document features in $\mathcal{R}[:, k]$; (3). the concatenation of the position and the average document features up to position $k$ (POS + PREDOC), as well as the proposed learnable state representations based on multi-head self-attention (ATTENTION). ORACLE here is used to show the gap from the upper bound. The results of our experiments are presented in Table 2 (with a full table including other click models is shown in Appendix B). For the PBM click model, it is evident that state embeddings utilizing position-based information, such as POS and POS+PREDOC, outperform other state embeddings. In contrast, for the CASCADE click model, state embeddings utilizing previous document features exhibit significantly stronger performance compared to those utilizing position information. Notably, our method, CUOLR, which dynamically learns the state embeddings during policy optimization, consistently achieves comparable performance compared to using hard-coded fixed state embeddings. This highlights the necessity of leveraging state representation in off-policy LTR and underscores the effectiveness of our proposed approach.

## 6 Conclusion

In this paper, we present an off-policy learning-to-rank formulation from the perspective of reinforcement learning. Our findings demonstrate that under this novel MDP formulation, RL algorithms can effectively address position bias and learn the optimal ranker for various click models, without the need for complex debiasing methods employed in unbiased learning to rank literature. This work establishes a direct connection between reinforcement learning and unbiased learning to rank through a concise MDP model. Specifically, we propose a novel off-policy learning-to-rank algorithm, CUOLR, which simultaneously learns efficient state representations and the optimal policy. Through empirical evaluation, we show that CUOLR achieves robust performance across a wide range of click models, consistently surpassing existing off-policy learning-to-rank methods tailored to those specific models. These compelling observations indicate that the extensive research conducted on offline

reinforcement learning can be leveraged for learning to rank with biased user feedback, opening up a promising new area for exploration.

## Acknowledgments and Disclosure of Funding

This work was supported in part by Google Cloud Research Credits Program. Mengdi Wang acknowledges the support by NSF grants DMS-1953686, IIS-2107304, CMMI-1653435, ONR grant 1006977, and C3.AI.

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

# A    Experiment Details

## A.1    Dataset Statistics

We conducted experiments on MSLR-WEB10K [4] and Yahoo! LETOR (set 1) [5] with semi-synthetic generated click data. Yahoo! LETOR comes from the Learn to Rank Challenge. It consists of 29,921 queries and 710K documents. Each query-document pair is represented by a 700-dimensional feature and annotated with a 5-level relevance label ranging from 0 to 4. MSLR-WEB10K dataset contains 10,000 queries and 125 retrieved documents on average. Each query-document pair is represented by a 136-dimensional feature vector and a 5-level relevance label. The dataset is partitioned into five parts with about the same number of queries, denoted as S1, S2, S3, S4, and S5, for five-fold cross-validation. All statistics of the used datasets are summarized in Table 3.

Table 3: Statistics of Learning to Rank Datasets

|  | # of queries | # of features | # of query-document pairs | rel level | # of folders |
|---|---|---|---|---|---|
| MSLR-WEB10K | 10000 | 136 | 1250k | 5 | 5 |
| Yahoo! LETOR | 29921 | 700 | 710k | 5 | 2 |

## A.2    Implementation Details

Before introducing the hyperparameters required for each algorithm, we first describe some global hyperparameters that are used commonly across all algorithms. We use batch size $B = 256$ queries per epoch, and use *nDCG@10* as the training objective for all baselines. We use *Adam* optimizer to train all the networks.

**Dual Learning Algorithm (DLA) [4].**    For DLA, two sub-models are being implemented: the *ranking (scoring) model* which is used to score each document; and the *propensity model* which is used to estimate the propensity for each document in the rank list. We use the MLP with two hidden layers of 256 units for both of them, and other hyperparameters are shown in Table 5.

**Inverse Propensity Weighting (IPW) [52, 26] & Cascade Model-based IPW (CM-IPW) [48].** For IPW and CM-IPW, we need to get the propensities from *Result Randomization*. In total, there are 10M random rank lists with different searching queries shown to the user (click model), and the parameters of each click model are estimated by *Maximize Likelihood Estimation (MLE)*. Estimation details are shown in Table 4, where $C_k$ and $C_{<k}$ indicate the click at rank $k$ and before rank $k$ respectively. The superscript $\cdot^{(s)}$ denotes the click of some click session $s$. Besides, we implement the ranking model the same way as that for DLA. Other hyperparameters are shown in Table 5.

**ORACLE.**    For ORACLE, we train a LambdaMART ranker Burges [6] with true labels on the training dataset and evaluate its performance on the test set. We leverage the *RankLib*[6] learning to rank library, and set the hyperparameters shown in Table 5. This utilizes the full-information data and serves as an upper bound of the performance for all algorithms utilizing the partial-information data, such as the generated clicks.

**CUOLR.**    For our algorithm CUOLR, there are three sets of hyperparameters used by the following sub-models: parameters for *state embedding model*, *RL policy*, as well as *RL critic*, where we use a 256-256 MLP to implement all the networks. Detailed hyperparameters are shown in Table 5.

# B    Additional Results

In this section, we present the complete results for all five click models (PBM, CASCADE, UBM, DCM, and CCM) on the two datasets: Yahoo! LETOR set 1 and MSLR-WEB10K. For the first two

---

[4] https://www.microsoft.com/en-us/research/project/mslr/

[5] https://webscope.sandbox.yahoo.com/

[6] https://sourceforge.net/p/lemur/wiki/RankLib/

Table 4: Details for IPW and CM-IPW.

| | PROPENSITY | PARAM. ESTIMATION |
|---|---|---|
| IPW | $P(E = 1 \mid k) = \theta_k$ | $\theta_k = \frac{\sum_{s \in \mathcal{S}} C_0^{(s)}}{\sum_{s \in \mathcal{S}} C_k^{(s)}}$ |
| CM-IPW | $P(E = 1 \mid k) \approx P(E = 1 \mid k, C_{<k})$ $= \Pi_{i<k}(1 - C_i(1 - \lambda_i))$ | $\lambda_k = 1 - \frac{\sum_{s \in \mathcal{S}_k} \mathcal{I}(C_{>k}^{(s)}=0)}{\sum_{s \in \mathcal{S}_k} 1}, \quad \mathcal{S}_k = \left\{ s : C_k^{(s)} = 1 \right\}$ |

Table 5: Hyperparameters for each algorithm used in the experiment.

| ALG | HYPERPARAMETERS |
|---|---|
| DLA | policy learning rate: 1e-4 propensity learning rate: 1e-4 loss type: softmax |
| IPW & CM-IPW | policy learning rate: 1e-4 loss type: softmax |
| ORACLE | number of trees: 1000 number of leaves for each tree: 100 shrinkage (learning rate): 0.01 min leaf support: 50 early stop: 100 |
| CUOLR | actor learning rate: 1e-4 critic learning rate: 1e-4 actor $\alpha$: 1e-10 (fixed) soft update $\tau$: 5e-3 discount $\gamma$: 0.8 |
| | embed learning rate: 1e-6 embed type: multi-head attention number of heads: 8 |
| | CQL $\alpha$: 1e-1 (fixed) |

studies in Section B.1 and B.2, we run 5 runs with different random seeds for the Yahoo! dataset. For the MSLR-WEB10K dataset which naturally comes with 5 folds, we take 1 run for each fold and aggregate the results. In the ablation experiment of conservatism for the offline RL algorithm in Section B.3, we only run 3 runs on Yahoo! due to the time limit. In all of our experiments, we use nDCG and ERR at positions 3,5,10 as evaluation metrics.

## B.1 Performance across Different Click Models

In this section, we present a comprehensive comparison between our proposed method, CUOLR, and various baseline approaches. Specifically, we examine the efficacy of DLA and IPW, which have been specifically designed for position-based models, as well as CM-IPS, which has been tailored for cascade-based models. The comparative results are presented in Table 7. In the case of position-based models such as PBM and UBM, it is evident that IPW demonstrates the most superior performance among all the considered baselines. Conversely, when evaluating cascade models such as cascade and DCM, the utilization of CM-IPW yields improved performance, as it takes into account the propensity estimation considering prior examinations and clicks. Among the diverse click models examined, our unified algorithm, CUOLR, consistently achieves the highest level of performance in terms of the ERR metrics across different positions. Furthermore, it consistently outperforms the other models in the majority of cases in terms of nDCG@10. This provides empirical verification of the effectiveness of our unified framework and the robustness of the CUOLR algorithm.

## B.2 State Representation Ablation Experiments

In this section, we present an ablation study focusing on the state embedding utilized in our algorithm, CUOLR. We compare the effectiveness of our proposed multi-head self-attention, augmented with positional embedding, against several heuristic hard-coded baselines for state embedding. These

Table 6: Performance comparison with different click models on Yahoo! LETOR set1. "*" and "**" indicate statistically significant improvement (p-value < 0.05 and p-value < 0.01 respectively) over the best baseline for each metric respectively.

| CLICK MODEL | ALG | ERR@3 | ERR@5 | ERR@10 | nDCG@3 | nDCG@5 | nDCG@10 |
|---|---|---|---|---|---|---|---|
| PBM | DLA | 0.418 | 0.439 | 0.455 | 0.669 | 0.691 | 0.742 |
| | CM-IPW | 0.418 | 0.440 | 0.456 | 0.669 | 0.692 | 0.743 |
| | IPW | 0.424 | 0.446 | 0.462 | **0.678** | **0.700** | 0.748 |
| | CUOLR(CQL) | **0.437**\*\* | **0.458**\*\* | **0.473**\*\* | 0.678 | **0.700** | **0.753**\* |
| | CUOLR(SAC) | **0.437**\*\* | **0.459**\*\* | **0.478**\*\* | 0.674 | **0.700** | **0.753**\*\* |
| CASCADE | DLA | 0.420 | 0.441 | 0.457 | 0.668 | 0.690 | 0.741 |
| | CM-IPW | 0.422 | 0.444 | 0.460 | **0.674** | **0.696** | 0.745 |
| | IPW | 0.420 | 0.442 | 0.457 | 0.668 | 0.690 | 0.740 |
| | CUOLR(CQL) | **0.436**\*\* | **0.459**\*\* | **0.479**\*\* | 0.668 | **0.696** | **0.748** |
| | CUOLR(SAC) | **0.440**\*\* | **0.461**\* | **0.478**\*\* | 0.670 | **0.696** | **0.748** |
| UBM | DLA | 0.426 | 0.448 | 0.463 | **0.687** | **0.708** | **0.756** |
| | CM-IPW | 0.415 | 0.437 | 0.453 | 0.664 | 0.689 | 0.740 |
| | IPW | 0.427 | 0.449 | 0.464 | 0.686 | **0.708** | **0.756** |
| | CUOLR(CQL) | **0.435**\*\* | **0.457**\*\* | **0.473**\*\* | 0.679 | 0.704 | **0.756** |
| | CUOLR(SAC) | **0.435**\*\* | **0.458**\*\* | **0.473**\*\* | 0.675 | 0.700 | 0.754 |
| DCM | DLA | 0.422 | 0.444 | 0.459 | 0.673 | 0.696 | 0.745 |
| | CM-IPW | 0.426 | 0.447 | 0.463 | **0.683** | **0.704** | 0.752 |
| | IPW | 0.422 | 0.444 | 0.459 | 0.674 | 0.697 | 0.746 |
| | CUOLR(CQL) | **0.437**\*\* | **0.461**\*\* | **0.474**\*\* | 0.676 | 0.699 | **0.753** |
| | CUOLR(SAC) | **0.436**\*\* | **0.461**\*\* | **0.475**\*\* | 0.675 | 0.703 | **0.755** |
| CCM | DLA | 0.427 | 0.449 | 0.464 | 0.685 | 0.707 | 0.754 |
| | CM-IPW | 0.416 | 0.438 | 0.453 | 0.659 | 0.684 | 0.736 |
| | IPW | 0.429 | 0.450 | 0.465 | **0.689** | **0.710** | 0.757 |
| | CUOLR(CQL) | **0.438**\*\* | **0.462**\*\* | **0.478**\*\* | 0.681 | 0.704 | **0.758** |
| | CUOLR(SAC) | **0.437**\*\* | **0.460**\*\* | **0.476**\*\* | 0.680 | 0.706 | **0.759** |
| | LOGGING | 0.359 | 0.385 | 0.403 | 0.595 | 0.630 | 0.693 |
| | ORACLE | **0.440** | **0.462** | **0.486** | **0.720** | **0.739** | **0.781** |

baselines include utilizing only positional information (POS), concatenating previous document information (PREDOC), and a combination of positional and document information (POS+PREDOC). The evaluation is performed on the Yahoo dataset, and the results are summarized in Table 8. Consistent with expectations, for position-based models (PBM), the most effective approach is utilizing only the positional information. Conversely, for cascade models (CASCADE), only considering the previous document information gives the best performance. In the case of more complicated models, such as DCM, CCM, and UBM, where the click model relies on both position and previous examinations, it becomes evident that incorporating a combination of positional information and previous document information yields the highest performance. Among all of them, it is worth to point out that our proposed state representation learning consistently attains comparable performance to the best baseline. Notably, our method possesses the advantage of automatically learning the optimal state representation, irrespective of the underlying assumptions of the click models.

### B.3 Conservatism for Offline RL Ablation Experiments

In this section, we conduct an ablation study to investigate the influence of the hyperparameter $\alpha$, which governs the conservatism, in the CQL algorithm. Specifically, we examine its effects on the Yahoo! dataset by varying $\alpha$ across a range of values: $\{0, 1e-3, 5e-3, 1e-2, 5e-2, 1e-1, 5e-1, 1e0, 5e0, 1e1, 5e1\}$. It is noteworthy that when $\alpha$ is set to 0, the CQL algorithm simplifies to the SAC algorithm [19]. Remarkably, we find that the performance of CUOLR remains consistently robust across the diverse $\alpha$ values, as long as they are not being restricted to be too close to the logging policy (i.e., large $\alpha$ values). This consistency underscores the effectiveness of our method, which demonstrates its ability to adapt to different underlying reinforcement learning (RL) algorithms. Interestingly, in contrast to classical offline RL datasets studied in Kumar et al. [29], where the conservatism parameter plays a substantial role, we observe that its impact is comparatively minor

Table 7: Performance comparison with different click models on MSLR-WEB10K. "*" and "**" indicate statistically significant improvement (p-value < 0.05 and p-value < 0.01 respectively) over the best baseline for each metric respectively.

| CLICK MODEL | ALG | ERR@3 | ERR@5 | ERR@10 | nDCG@3 | nDCG@5 | nDCG@10 |
|---|---|---|---|---|---|---|---|
| PBM | DLA | 0.230 | 0.256 | 0.278 | 0.343 | 0.356 | 0.384 |
|  | CM-IPW | 0.229 | 0.255 | 0.277 | 0.341 | 0.354 | 0.383 |
|  | IPW | 0.257 | 0.281 | 0.301 | 0.356 | 0.367 | 0.390 |
|  | CUOLR(CQL) | **0.257** | **0.281** | **0.303** | **0.369**\*\* | **0.380**\*\* | **0.406**\*\* |
|  | CUOLR(SAC) | 0.255 | 0.279 | **0.301** | **0.368**\*\* | **0.379**\*\* | **0.404**\*\* |
| CASCADE | DLA | 0.239 | 0.265 | 0.286 | 0.354 | 0.365 | 0.392 |
|  | CM-IPW | **0.259** | **0.283** | **0.304** | 0.367 | 0.378 | 0.403 |
|  | IPW | 0.232 | 0.257 | 0.279 | 0.347 | 0.358 | 0.387 |
|  | CUOLR(CQL) | 0.255 | 0.280 | 0.301 | 0.366 | **0.378** | **0.404** |
|  | CUOLR(SAC) | 0.255 | 0.279 | 0.301 | **0.368**\* | **0.379** | **0.405**\* |
| UBM | DLA | 0.257 | 0.280 | 0.300 | 0.369 | 0.377 | 0.400 |
|  | CM-IPW | 0.253 | 0.276 | 0.297 | 0.354 | 0.362 | 0.386 |
|  | IPW | 0.258 | 0.281 | 0.301 | 0.359 | 0.367 | 0.390 |
|  | CUOLR(CQL) | 0.255 | 0.280 | 0.301 | **0.373**\*\* | **0.384**\*\* | **0.408**\*\* |
|  | CUOLR(SAC) | **0.260** | **0.284** | **0.306**\* | **0.374**\* | **0.384**\* | **0.408**\* |
| DCM | DLA | 0.254 | 0.279 | 0.299 | 0.368 | 0.378 | 0.404 |
|  | CM-IPW | **0.259** | **0.284** | **0.304** | 0.367 | 0.379 | 0.402 |
|  | IPW | 0.256 | 0.278 | 0.299 | 0.354 | 0.363 | 0.387 |
|  | CUOLR(CQL) | 0.254 | 0.278 | 0.300 | **0.369** | **0.380** | **0.405** |
|  | CUOLR(SAC) | 0.254 | 0.278 | 0.300 | 0.367 | 0.378 | 0.403 |
| CCM | DLA | 0.252 | 0.275 | 0.295 | 0.347 | 0.356 | 0.379 |
|  | CM-IPW | 0.227 | 0.249 | 0.270 | 0.305 | 0.316 | 0.342 |
|  | IPW | **0.255** | 0.278 | 0.298 | 0.351 | 0.360 | 0.383 |
|  | CUOLR(CQL) | **0.255** | **0.280** | **0.301**\* | **0.373**\*\* | **0.384**\*\* | **0.408**\*\* |
|  | CUOLR(SAC) | **0.260**\* | **0.284**\*\* | **0.305**\*\* | **0.374**\*\* | **0.383**\*\* | **0.408**\*\* |
|  | LOGGING | 0.180 | 0.206 | 0.230 | 0.288 | 0.304 | 0.338 |
|  | ORACLE | **0.305** | **0.328** | **0.347** | **0.426** | **0.432** | **0.453** |

in the offline learning to rank dataset. This observation is worth further investigation for better understanding the impact of conservatism in offline LTR settings.

Besides, To figure out the effect of conservatism, we do experiments on several click models on Yahoo! dataset to compare the performance between simple SAC and CQL with the optimal conservative parameter $\alpha$. The optimal $\alpha$ is selected through grid search ranging from $\{1e-3, 5e-3, 1e-2, 5e-2, 1e-1, 5e-1, 1e0, 5e0, 1e1, 5e1\}$. The results are shown in Table 9. It's obvious that the performance of CQL with optimal $\alpha$ is consistently better than that of CQL, especially in the NDCG metric. This improvement further illustrates the necessity of conservatism in Off-policy learning to rank task.

## B.4    Data Quality for Offline Reinforcement Learning to Rank

In this section, we conduct an ablation study to clarify the effect of data quality to the offline RL algorithms. In usual cases, data quality is important to most offline algorithms, including the offline RL algorithm. To verify the effect of data quality, we conduct experiments on the Web10k dataset with different offline data. Specifically, on each click model, the logging policy is trained with different portions of train data: 1) SVMRank trained with 1% train data; 2) SVMRank trained with 0.01% train data; 3) random policy without pre-training. Then the quality of the train data gathered by the three logging policy decreases, and the performance of CQL on them is displayed in Table 10. We find that the algorithm trained with data in better quality (first line for each click model) performs significantly better than the other two. Besides, it's interesting that the algorithm trained with random logging data can't even beat the logging policy trained with 1% train data, which further highlight the significance of data quality.

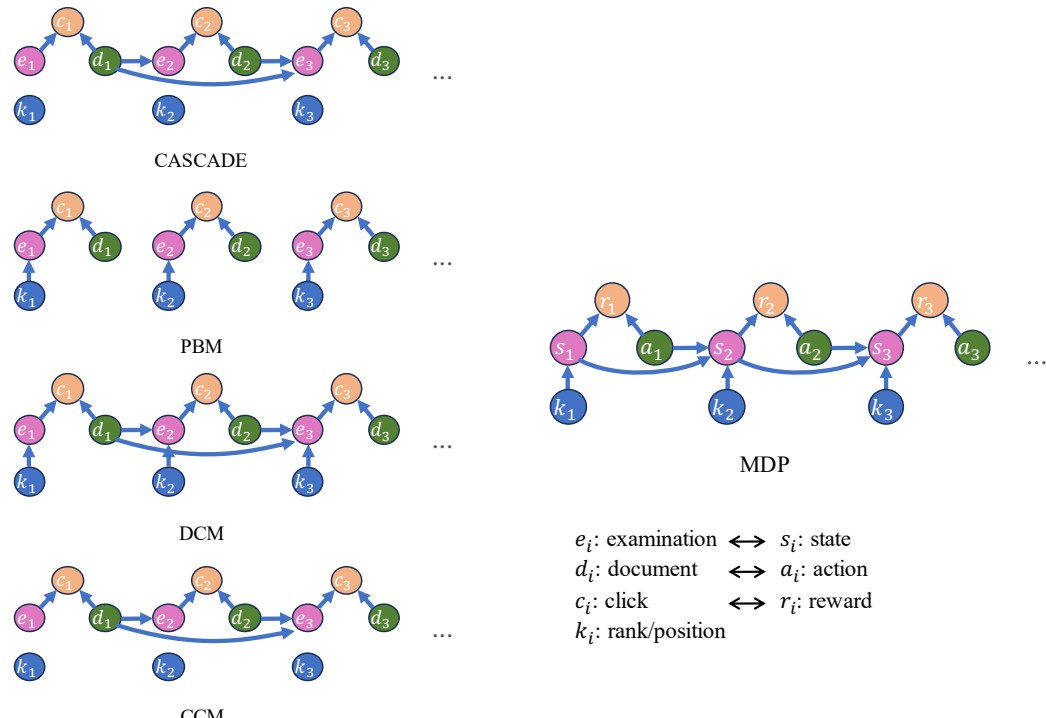

Figure 1: Graphical models of different click models and our MDP formulation about the learning to rank problem.

## C  Click Model and MDP Formulation

In this section, we present a comprehensive overview of different click models employed in the paper, namely PBM, CASCADE, DCM, CCM, and UBM. Additionally, we demonstrate the graphical models associated with each click model and how they could be unified into the Markov Decision Process (MDP) framework, as depicted in Figure 1.

**PBM [24].**   The position-based model is a model where the probability of clicking on an item depends on both its identity and rank. The examination probability is rank-dependent only, i.e.,

$$\chi(\mathcal{R}, k) = \chi(k).$$

**CASCADE [12].**   The cascade model assumes that the user scans a rank list $\mathcal{R}$ from top to bottom. If a document at rank $k$ is examined and clicked, the user stops browsing the remaining documents. Otherwise, the user goes on to the next rank $k + 1$ with probability one. The first document $d_1$ is always examined. The document at rank $k$ will be examined if and only if the previous $k - 1$ documents are not clicked. Therefore we have:

$$\chi(\mathcal{R}, k) = \Pi_{i=1}^{k-1} \left(1 - \alpha(\mathcal{R}(i))\right).$$

**DCM [48, 18].**   The dependent click model assumes that the user examines the results from top to bottom until an attractive result is found, $P(E_{k+1} = 1 \mid E_k = 1, C_k = 0) = 1$, where $E_k$ is the examination indicator at rank $k$. After each click, there is a rank-dependent chance that the user is unsatisfied, $P(E_{k+1} = 1 \mid C_k = 1) = \lambda_k$. Therefore, we have:

$$\chi(\mathcal{R}, k) = \Pi_{i=1}^{k-1} \left(1 - \alpha(\mathcal{R}(i)) \cdot (1 - \lambda_i)\right).$$

**CCM [17].**   The click chain model (CCM) is a generalization of the dependent click model where continuing to examine the results before a click is not deterministic, i.e. $P(E_{j+1} = 1 \mid E_j = $

Table 8: Performance of CQL algorithm with different state embedding. The experiments are conducted on Yahoo! set 1 and click models are PBM, CASCADE, UBM, DCM, and CCM

| CLICK MODEL | STATE EMBED | ERR@3 | ERR@5 | ERR@10 | nDCG@3 | nDCG@5 | nDCG@10 |
|---|---|---|---|---|---|---|---|
| PBM | POS | 0.417 | 0.439 | 0.455 | 0.668 | 0.691 | 0.742 |
| | PREDOC | 0.413 | 0.435 | 0.451 | 0.657 | 0.682 | 0.734 |
| | POS+PREDOC | 0.418 | 0.460 | 0.456 | 0.663 | 0.685 | 0.734 |
| | ATTENTION | **0.437** | **0.459** | **0.478** | **0.674** | **0.700** | **0.753** |
| CASCADE | POS | 0.404 | 0.426 | 0.442 | 0.646 | 0.668 | 0.724 |
| | PREDOC | 0.434 | 0.456 | 0.472 | **0.679** | **0.703** | **0.754** |
| | POS+PREDOC | 0.432 | 0.453 | 0.470 | 0.664 | 0.686 | 0.740 |
| | ATTENTION | **0.440** | **0.461** | **0.478** | 0.669 | 0.696 | 0.748 |
| UBM | POS | 0.420 | 0.442 | 0.458 | 0.678 | 0.700 | 0.747 |
| | PREDOC | 0.410 | 0.433 | 0.449 | 0.666 | 0.691 | 0.743 |
| | POS+PREDOC | 0.416 | 0.439 | 0.454 | 0.676 | 0.700 | 0.750 |
| | ATTENTION | **0.435** | **0.457** | **0.473** | **0.679** | **0.704** | **0.756** |
| DCM | POS | 0.415 | 0.437 | 0.453 | 0.658 | 0.680 | 0.731 |
| | PREDOC | 0.435 | 0.457 | 0.473 | 0.682 | 0.706 | **0.758** |
| | POS+PREDOC | **0.439** | **0.461** | **0.477** | **0.687** | **0.709** | **0.758** |
| | ATTENTION | 0.437 | **0.461** | 0.475 | 0.675 | 0.703 | 0.755 |
| CCM | POS | 0.421 | 0.443 | 0.458 | 0.678 | 0.701 | 0.748 |
| | PREDOC | 0.420 | 0.443 | 0.459 | 0.677 | 0.702 | 0.754 |
| | POS+PREDOC | 0.426 | 0.448 | 0.464 | **0.685** | **0.710** | **0.759** |
| | ATTENTION | **0.437** | **0.460** | **0.476** | 0.680 | 0.706 | **0.759** |
| | ORACLE | **0.440** | **0.462** | **0.486** | **0.720** | **0.739** | **0.781** |

Table 9: Comparison of SAC and CQL with optimal alpha on Yahoo. To show the impact of the offline algorithm in off-policy learning to rank, we compare SAC and CQL with different degrees of conservatism. We control the degree of conservatism via the parameter $\alpha$ (See details in Eq.5 in the main paper). The $\alpha$s range from $\{1e-3, 5e-3, 1e-2, 5e-2, 1e-1, 5e-1, 1e0, 5e0, 1e1, 5e1\}$ and we do grid search to find the optimal $\alpha$.

| CLICK MODEL | ALG | ERR@K | | | NDCG@K | | |
|---|---|---|---|---|---|---|---|
| | | K=3 | K=5 | K=10 | K=3 | K=5 | K=10 |
| PBM | SAC | 0.437 | 0.459 | 0.478 | 0.674 | 0.700 | 0.753 |
| | CQL(optimal $\alpha$) | 0.437 | 0.460 | 0.478 | 0.692 | 0.715 | 0.766 |
| CASCADE | SAC | 0.440 | 0.461 | 0.478 | 0.670 | 0.696 | 0.748 |
| | CQL(optimal $\alpha$) | 0.440 | 0.461 | 0.479 | 0.691 | 0.715 | 0.765 |
| DCM | SAC | 0.436 | 0.461 | 0.475 | 0.675 | 0.703 | 0.755 |
| | CQL(optimal $\alpha$) | 0.439 | 0.461 | 0.477 | 0.694 | 0.718 | 0.767 |
| CCM | SAC | 0.437 | 0.460 | 0.476 | 0.680 | 0.706 | 0.759 |
| | CQL(optimal $\alpha$) | 0.438 | 0.462 | 0.479 | 0.691 | 0.712 | 0.766 |

$1, C_j = 0) = \alpha_1$. The probability of continuing after a click is not position dependent, but relevance dependent, $P(E_{j+1} \mid C_j = 1) = \alpha_2(1 - R_i) + \alpha_3 R_i$, where $R_i$ is the relevance of the $i^{th}$ document in the rank list. Therefore, the examination probability at each position can be written as:

$$\chi(\mathcal{R}, k) = \Pi_{i=1}^{k-1}(1 - \alpha(\mathcal{R}(i)) \cdot (1 - \alpha_2(1 - \mathcal{R}(i)) - \alpha_3 \mathcal{R}(i))$$

**UBM [13].** The user browsing model (UBM) is an extension of the PBM model with some elements of the cascade model. The whole model is position-based, but for the examination probability, it considers previous clicks. Specifically, the examination probability depends not only on the rank of the document $k$, but also on the rank of the previously clicked document $k'$, which is modeled by a set of parameters $\gamma_{kk'}$, i.e. $P(E_k = 1 \mid C_1 = c_1, \ldots, C_{k-1} = c_{k-1}) = \gamma_{kk'}$, where $k'$ is the rank of the previous clicked document or 0 if none of them was clicked, i.e. $k' = \max\{r \in \{0, \ldots, k-1\}\}$ :

Table 10: Comparison of different logging policies on Web10k. On each click model, we compare three different logging policies: SVMRank trained with 1% train data, SVMRank trained with 0.01% train data, and random policy.

| CLICK MODEL | LOGGING | ERR@K | | | NDCG@K | | |
|---|---|---|---|---|---|---|---|
| | | K=3 | K=5 | K=10 | K=3 | K=5 | K=10 |
| PBM | por=1% | 0.257 | 0.281 | 0.303 | 0.369 | 0.380 | 0.406 |
| | por=0.01% | 0.214 | 0.239 | 0.262 | 0.324 | 0.339 | 0.367 |
| | rand | 0.111 | 0.130 | 0.151 | 0.159 | 0.175 | 0.201 |
| CASCADE | por=1% | 0.255 | 0.280 | 0.301 | 0.366 | 0.378 | 0.404 |
| | por=0.01% | 0.214 | 0.239 | 0.262 | 0.326 | 0.340 | 0.368 |
| | rand | 0.117 | 0.136 | 0.157 | 0.163 | 0.180 | 0.208 |
| DCM | por=1% | 0.254 | 0.278 | 0.300 | 0.369 | 0.380 | 0.405 |
| | por=0.01% | 0.212 | 0.238 | 0.260 | 0.324 | 0.338 | 0.367 |
| | rand | 0.121 | 0.139 | 0.159 | 0.167 | 0.178 | 0.204 |
| CCM | por=1% | 0.255 | 0.280 | 0.301 | 0.373 | 0.384 | 0.408 |
| | por=0.01% | 0.214 | 0.239 | 0.261 | 0.324 | 0.339 | 0.367 |
| | rand | 0.123 | 0.142 | 0.162 | 0.163 | 0.179 | 0.208 |

$c_r = 1$

$$\chi(\mathcal{R}, k) = \gamma_{kk'}$$

Zoghi et al. [63] showed that PBM and CM satisfied Assumption 3.1 where optimal ranking list leads to optimal total clicks. In our future work, it would be interesting to show that other click models also satisfy this assumption.

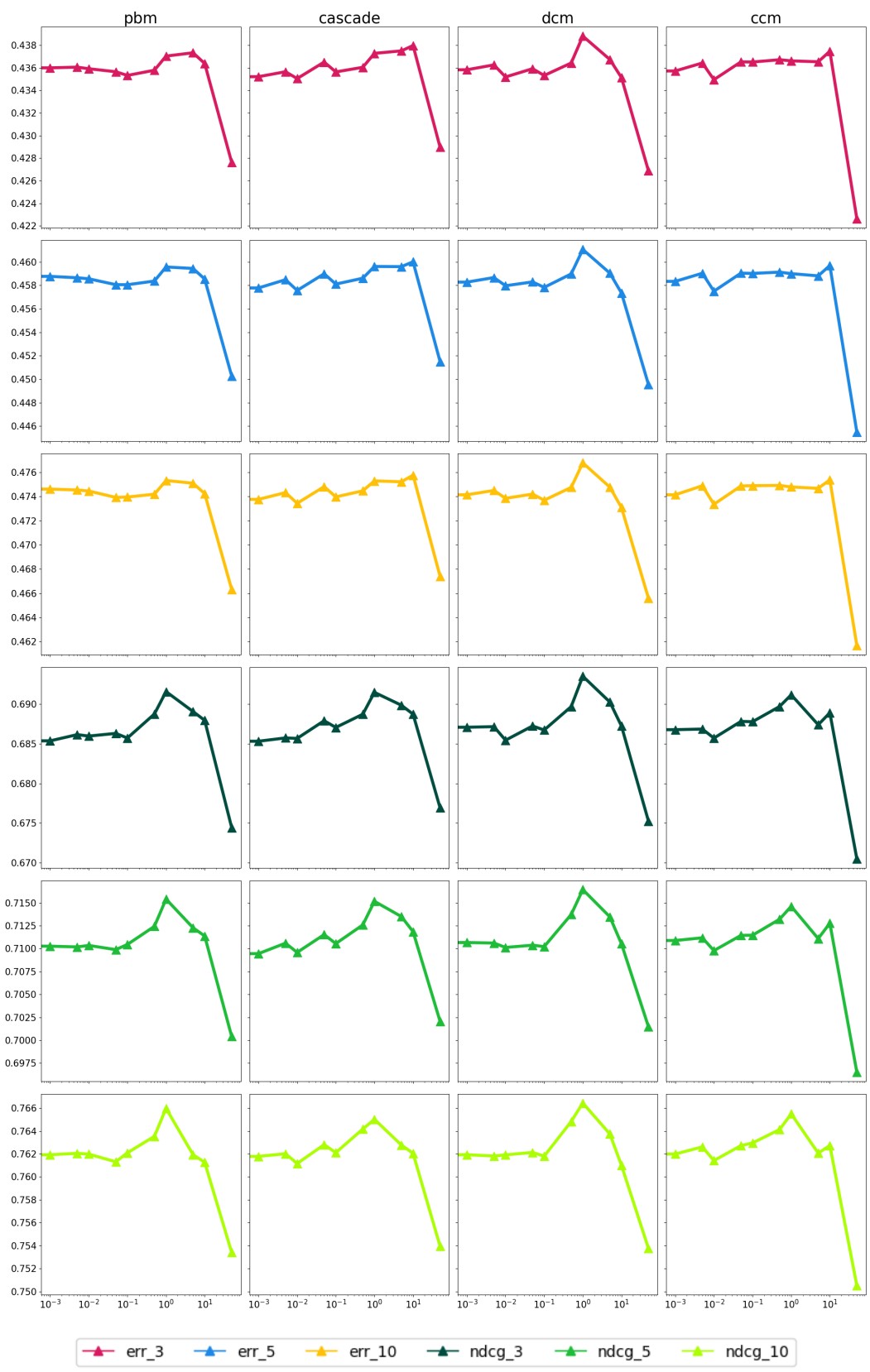

Figure 2: NDCG@10 and ERR@10 of CUOLR (with CQL) on Yahoo! dataset set 1, with different conservative parameters $\alpha$s. Each row is the performance of a metric (ERR, NDCG@3,5,10) on the four click models (PBM, CASCADE, DCM, CCM).

