# OpenReview forum: "Unified Off-Policy Learning to Rank: a Reinforcement Learning Perspective"
_NeurIPS.cc/2023/Conference — NeurIPS 2023 poster_

### Official Review · Reviewer_ynTd · 2023-07-03

**Soundness:** 3 good
**Presentation:** 2 fair
**Contribution:** 2 fair
**Rating:** 5
**Confidence:** 3

**Summary:**

This paper aims to unify the assumptions on user behavior in a ranking called click model by formulating the ranking problem as a click-model agnostic Markov Decision Process (MDP). By doing so, the paper proposes to reduce the Off-Policy Learning to Rank (LTR) problem to a variant of offline RL, which does not require precise estimation of the click models. Under this formulation, the proposed offline RL method, which incorporates state representation learning to the well-known Conservative Q-Learning (CQL), enables more accurate estimations of ranking metrics than baseline estimators in real-world datasets.

**Strengths:**

1. The manuscript is easy to follow. The motivation for unifying click models and introducing the offline RL framework is clearly explained.

2. The RL formulation of LTR is reasonable, and showing the connection between offline RL and Off-Policy LTR would be insightful for the LTR community.

3. The ablation study on state representation learning is insightful.


**Weaknesses:**

1. While the connection between LTR and offline RL is interesting, the proposed method (CUOLR) itself is not a fundamentally new framework for offline RL. In particular, the proposed state representation learning method, which applies positional encoding and attention to the inputs, seems to be an engineering effort rather than a very novel framework.

2. In experiments, an offline RL algorithm (CQL) does not show advantages over a simple RL baseline (SAC). While I acknowledge the author(s)’ contribution to formulating LTR as an RL problem, I think the algorithm has room for improvement.


**Questions:**

- In general, SAC does not perform very well in the offline RL setting and thus the offline RL algorithms, including CQL, have been proposed. However, the experiment results show that CUOLR (CQL) and CUOLR (SAC) perform competitively. Could you provide some justification for this result?

**Limitations:**

This limitation is not special for this paper, but classical click models and MDP both assume that the reward observed at each position are not affected by lower positions including the neighboring ones. If this assumption does not hold, LTR may introduce some bias and offline RL may also ignore some causal relation between actions and rewards.

---

> ### Author Rebuttal · Authors · 2023-08-10
>
> We appreciate the reviewer for recognizing the advantages of our unified RL formulation, connecting LTR and RL solution, and our ablation study.
>
> **Q:** The proposed method (CUOLR) itself is not a fundamentally new framework for offline RL, and the proposed state representation learning method seems to be an engineering effort rather than a very novel framework.
>
> **A:** While applying positional encoding and attention to the inputs contains certain engineering efforts, we would like to emphasize that our proposed CUOLR method is novel in terms of its unified MDP formulation that is agnostic to various click models, novel state representation design and flexible to apply any offline RL solvers. This also gives a much better performance than the tailored method for each specific click model.
>
> **Q:** Classical click models and MDP both assume that the reward observed at each position are not affected by lower positions including the neighboring ones. If this assumption does not hold, LTR may introduce some bias and offline RL may also ignore some causal relation between actions and rewards.
>
> **A:** We agree with the reviewer that classical click models assume that the reward is not affected by lower positions. It would be a challenging yet interesting future work to broaden the scope and relax the assumption.

---

### Official Review · Reviewer_79J3 · 2023-07-06

**Soundness:** 3 good
**Presentation:** 3 good
**Contribution:** 3 good
**Rating:** 6
**Confidence:** 4

**Summary:**

This paper presents a unified approach for off-policy learning to rank (LTR) that is adaptable to general click models. The authors formulate off-policy LTR as a Markov Decision Process (MDP) and leverage offline reinforcement learning (RL) techniques to optimize the ranking process. They propose the Click Model-Agnostic Unified Off-policy Learning to Rank (CUOLR) method, which can be easily applied to a wide range of click models. The authors provide empirical evidence demonstrating the effectiveness of CUOLR in comparison to state-of-the-art off-policy LTR algorithms.


**Strengths:**

● The paper presents an innovative and practical methodology for off-policy learning to rank. The formulation of off-policy LTR as an MDP and the use of offline RL techniques provide a comprehensive and adaptable approach for ranking optimization.
● The authors provide insightful empirical findings, showing that the CUOLR method consistently outperforms state-of-the-art off-policy learning to rank algorithms. The results on various large-scale datasets demonstrate the effectiveness, consistency, and robustness of CUOLR under different click models.
● The paper presents a detailed introduction on state representation learning. There are also empirical analyses on its practical effect.

**Weaknesses:**

● The synthetic dataset and the offline evaluation can give biased evaluation results of the algorithms.
● There are some related works that are not mentioned in this paper. Cai, et al. [1] also apply RL techniques to recommendation systems and provide a new MDP formulation. It also includes ranking scores in the formulation that can be regarded as a downstream application to this paper. Xue, et al. [2] provides another MDP formulation of RL for optimizing long-term user engagement.
● There remain some issues unsolved in the paper. See the questions for details.

[1] Cai, Qingpeng, et al. "Reinforcing User Retention in a Billion Scale Short Video Recommender
System." arXiv preprint arXiv:2302.01724 (2023).
[2] Xue, Wanqi, et al. "PrefRec: Preference-based Recommender Systems for Reinforcing Long-term User Engagement." arXiv preprint arXiv:2212.02779 (2022).


**Questions:**

● Lines130-133 are hard to understand. Why not include remaining documents that are yet to rank? How will the policy perform on $s_0$, where nothing will be included as input?
● The introduction of dynamic action space can introduce dynamics shift in RL and unstable training. Are there any specific techniques to handle this issue?
● Why does SAC obtain similar performance with CQL when learning in static datasets? Will the issue of policy distribution mismatch lead to poor performance?

**Limitations:**

The formulation and evaluation are limited to off-policy/offline RL that learn from a static dataset. In applications where there are adequate online interaction data, it will be helpful to consider some online RL counterparts.

---

> ### Author Rebuttal · Authors · 2023-08-10
>
> We thank the reviewer for the positive comments on our work.
>
> **Q:** The synthetic dataset and the offline evaluation can give biased evaluation results of the algorithms.
>
> **A:**  First, same as previous off-policy learning to rank research (consistent with all baseline methods), we conduct the semi-synthetic experiments where the training data are all the direct feedback from a "simulated" user (the click model reflects his/her behavior) on real-world queries. This is different from offline evaluation since we have control of the environment and know the groundtruth of any ranking list (trajectory).  In this setup, the only bias comes from the click model, which is different from the bias in typical offline evaluation RL. For more details, please see the experiment setup section (section 5.1).
>
> **Q:** There are some related works that are not mentioned in this paper. Cai, et al. [1] also apply RL techniques to recommendation systems and provide a new MDP formulation. It also includes ranking scores in the formulation that can be regarded as a downstream application to this paper. Xue, et al. [2] provides another MDP formulation of RL for optimizing long-term user engagement.
> ([1] Cai, Qingpeng, et al. "Reinforcing User Retention in a Billion Scale Short Video Recommender System." arXiv preprint arXiv:2302.01724 (2023). [2] Xue, Wanqi, et al. "PrefRec: Preference-based Recommender Systems for Reinforcing Long-term User Engagement." arXiv preprint arXiv:2212.02779 (2022).)
>
> **A:** Thank you for providing these two papers on RL for recommendation. Their settings are different from our LTR setup, we will definitely add the discussion in our next version.
>
> **Q:** Lines130-133 are hard to understand. Why not include remaining documents that are yet to rank? How will the policy perform on s0, where nothing will be included as input?
>
> **A:** According to click models, the click at each position only depends on current position and previous documents. Remaining documents that are yet to rank are considered in action set instead of our state representation. At position 0 (state $s_0$), the policy will choose the document with the highest relevance/attractiveness from all candidates.
>
> **Q:** The introduction of dynamic action space can introduce dynamics shift in RL and unstable training. Are there any specific techniques to handle this issue?
>
> **A:** Thank you for pointing out this question. We agree that the raw action space is dynamic. However, as we utilize the attention model and project the raw action space into a latent action space, there is intended to be less dynamic shift happening there.
>
> **Q:** Why does SAC obtain similar performance with CQL when learning in static datasets? Will the issue of policy distribution mismatch lead to poor performance?
>
> **A:** Please refer to our answer in the general response to all reviewers regarding the performance difference between SAC and CQL.

---

> > ### Author Response · Authors · 2023-08-15
> >
> > Dear Reviewer79J3,
> >
> > We were wondering if you have gotten a chance to go through our responses and the additional experiments we add to the paper, and if these revisions and responses address your concerns regarding the paper. We are happy to address any remaining concerns and would really appreciate it if you engage in a discussion with us.
> >
> > Thank you so much!

---

> > > ### Comment · Reviewer_79J3 · 2023-08-16
> > > **Reply**
> > >
> > > Thanks for the rebuttal, I am more pleased with the paper. I will keep my original score and wish you good luck.

---

> > > > ### Author Response · Authors · 2023-08-21
> > > > **Thank you**
> > > >
> > > > We sincerely thank the reviewer for the time and effort in reviewing our paper.
> > > >
> > > >
> > > > Regards,
> > > >
> > > > Authors

---

### Official Review · Reviewer_atvb · 2023-07-07

**Soundness:** 2 fair
**Presentation:** 2 fair
**Contribution:** 2 fair
**Rating:** 6
**Confidence:** 2

**Summary:**

The paper talks about how to model user behaviors with positional biases in an online search system. The paper proposes a unified RL framework to generalize three common types of positional bias models: Position-Based Modeling (PBM), CASCADE (each click depends on the previous click), and Dependent Click Models (DCM). By capturing all previous clicks into a state, the paper suggests that standard offline RL algorithms, such as Conservative Q-Learning (CQL) and Soft Actor Critic (SAC), can be used to estimate and optimize for the positional-aware ranking policies. Simulation experiments are included to support the claims.

**Strengths:**

Originality. The paper observes commonalities in three different types of methods and proposes a unified method to combine them. The generality of the proposal is further examined by simulating different cases from each of the method and showing similar performance using the proposed method. The observation and the empirical validation feel original to me.

The authors addressed my concerns and I have adjusted my scores accordingly.

**Weaknesses:**

Significance. The paper made a rather limiting assumption that all users in the same search system follow a single pattern of positional bias. This may not be true, as different users may have different positional biases. A more practical approach is to insert perturbations to the ranking positions to estimate the true positional effects using IPS. To mitigate the risks of perturbations, methods have been introduced to swap only adjacent search results. Please see this paper as an example: https://www.cs.cornell.edu/people/tj/publications/agarwal_etal_19a.pdf

Clarity. The description of the proposed method lacks sufficient details (see additional questions). Also the experimental results (Table 1) did not seem to contain significant differences between the algorithms being considered. This leads me to wonder if the proposed RL solution is actually easy to find, or is it fundamentally difficult to implement and the results may be sensitive to hyperparameter choices. The paper did not discuss common limitations of RL algorithms.

**Questions:**

Table 1. The table compares the proposed method with baseline methods, one of which is used to simulate the ground-truth user behaviors. However, in the results, the baseline method which was used to simulate the user behaviors did not perform the best in estimating the user behaviors. Is this expected? Also, the differences between all methods seem pretty small.

Algorithm 1. Line 6, where is psi defined?

Eq (5). What is the difference between CQL and SAC? Why is there only one equation being presented for both algorithms? More details would be appreciated for clarity purposes.

**Limitations:**

No, the paper did not discuss the choice of hyperparameters or the closeness of the experimental results. They seem to be common challenges with RL algorithms.

---

> ### Author Rebuttal · Authors · 2023-08-10
>
> We would like to thank the reviewer for recognizing the originality of our unified formulation and solution.
>
> **Q:** The paper made a rather limiting assumption that all users in the same search system follow a single pattern of positional bias. This may not be true, as different users may have different positional biases. A more practical approach is to insert perturbations to the ranking positions to estimate the true positional effects using IPS. To mitigate the risks of perturbations, methods have been introduced to swap only adjacent search results.
>
> **A:** We thank the reviewer for pointing out the reference on the estimation of propensities and it is worth pointing out that in the reference paper, the probability of being examined actually only depends on the rank, and it is the same across users. Instead, we want to point out that one of our main advantages is that we do not need to learn a separate propensity estimation model.
>
> **Q:** The description of the proposed method lacks sufficient details. Also the experimental results (Table 1) did not seem to contain significant differences between the algorithms being considered. The paper did not discuss common limitations of RL algorithms.
>
> **A:** We provide details of our solution using SAC and CQL and their hyper-parameters in the Appendix (see Table 5 in Section B for details). Please refer to our answer in the general response to all reviewers regarding performance difference between SAC and CQL.
>
> **Q:** In the results, the baseline method which was used to simulate the user behaviors did not perform the best in estimating the user behaviors. Is this expected? Also, the differences between all methods seem pretty small.
>
> **A:** If we understand correctly that "the baseline method which was used to simulate the user behaviors" refers to IPW/DLA in PBM model and CM-IPW in Cascade model in Table 1, we would like to clarify that it does not have to perform best. Note that while designed for the specific click models, these methods still need to estimate parameters such as attractiveness and/or bias parameters in click model, and whether they can learn efficiently determines their performance. But we can see that methods designed for specific click model generally perform better than mismatched methods, e.g., CM-IPW>IPW/DLA in Cascade model.
>
> **Q:** Algorithm 1. Line 6, where is psi defined?
>
> **A:** We apologize for the confusion. $\psi$ is but mentioned in Line 2, the suffix of $\phi_\psi(\cdot,\cdot)$. It refers to the trainable parameters in the embedding model $\phi$.
>
> **Q:** Eq (5). What is the difference between CQL and SAC?
>
> **A:** According to Eq.5 in Line 244, the CQL algorithm overcomes the distribution shift problem in offline RL by Conservative Q-learning. In its loss function, a conservative term is added (the first term in the RHS of the equation after $\alpha$) to constrain the difference between the logging policy and our trained policy in case of the overestimation problem. More details can be found in [1] and we will add more discussion in next version.
>
> [1] Aviral Kumar, Aurick Zhou, George Tucker, and Sergey Levine. Conservative q-learning for offline reinforcement learning. Advances in Neural Information Processing Systems, 33:1179–1191, 2020*

---

> > ### Author Response · Authors · 2023-08-16
> >
> > Dear Reviewer atvb,
> >
> > Thanks for your review! We were wondering if you have gotten a chance to go through our responses and the additional experiments we add to the paper, and if these revisions and responses address your concerns regarding the paper. We are happy to address any remaining concerns and would really appreciate it if you engage in a discussion with us.
> >
> > Thank you so much!

---

> > > ### Comment · Reviewer_atvb · 2023-08-20
> > > **Thanks**
> > >
> > > The authors addressed my concerns and I have adjusted my scores accordingly.
> > >
> > > Regarding Q1, my original question was regarding perturbation during data logging/generation. I am from a recommendation background where such perturbations are often necessary. For search problems, the queries may already contain enough randomness such that further perturbations may not be necessary. The authors should briefly discuss these details, if they apply.

---

> > > > ### Author Response · Authors · 2023-08-21
> > > > **Thank you for the follow-up**
> > > >
> > > > Thank you for the discussion and for adjusting the score. We appreciate your efforts in reviewing our paper. Regarding Q1, while the perturbations have not been applied to the baselines and our method, we will discuss the suggested paper in the next version.

---

### Official Review · Reviewer_fhCJ · 2023-07-11

**Soundness:** 2 fair
**Presentation:** 3 good
**Contribution:** 3 good
**Rating:** 6
**Confidence:** 2

**Summary:**

- Authors propose a unified off-policy reinforcement learning-based approach for learning to rank (LTR) problem that is adaptable to all general click models.
- Authors argue that a user's behavior in terms of clicks can be modeled by a Markov Decision Process (MDP), thereby allowing them to look at LTR from the perspective of offline reinforcement learning.
- Authors' key contribution is a novel state representation that takes into account features of all the items/documents presented to a user. Specifically, they represent state at position $k$ by $s_k = [(d_1, d_2, \cdots, d_{k-1}), k]$ to capture the context of user behavior prior to reaching position $k$-ranked position. The action set at position $k$ is all the available items/documents that haven't been presented to the user in prior $k-1$ positions, and the reward is simply the users' click behavior when presented with action $a_k$.
- Authors use positional encoding and self-attention layer to extract the state representation given the sequence of items/documents at prior ranked positions, i.e., $(d_1, d_2, \cdots, d_{k-1})$. They train an end-to-end model that jointly optimizes for state representation as well as the policy.

**Strengths:**

**Motivation**
- Authors present a strong motivation for their approach based upon the need for a unified a click model-agnostic LTR method which generalizes well to use of any RL algorithm.
- Authors do a good job at placing their work in context of related work in the field by comparing/contrasting their approach with other works.
_Note_: I am not updated on the state-of-the-art RL approaches for LTR, and therefore may not be aware of some recent works.

**Technical Presentation**
- Authors do a good job at formalizing the problem and providing all the relevant technical information in Sections 3 and 4 that would be required for reproducibility.
- Authors contribution is quite easy and intuitive to understand.

**Experiments**
- Authors evaluate their approach (using CQL and SAC) in comparison to baselines such as Dual Learning Algorithm (DLA), Inverse Propensity Weighting (IPW) algorithm and Cascade-model based IPW, as well as LambdaMart model which serves an upper bound. The metrics of authors' proposed approach are significantly favorable in comparison to baselines. _Note_: Just as above, because I am unaware of the SOTA, I am unable to comment on whether the right baselines have been used for comparison.
- Authors provide ablation study to validate the need for each component of their proposed approach.

**Weaknesses:**

**Is this a Markov Decision Process?**

Formally, the Markovian property of MDP refers to the fact that given states $s_i$ and actions $a_i$, the equality $\mathrm{Pr}(s_t | s_0, a_0, s_1, a_1, \cdots, s_{t-1}, a_{t-1}) = \mathrm{Pr}(s_t | s_{t-1}, a_{t-1})$ holds. Intuitively, it means that the probability of landing in state $s_t$ of the Markov chain only depends on the last state-action pair $(s_{t-1}, a_{t-1})$, and not the state-action pairs preceding that.

Given the authors' approach uses state representation as $s_k = [(d_1, d_2, \cdots, d_{k-1}), k]$, it is unclear to me how/if the Markovian property still hold?

**Effect of Click Data Generation**

Authors mention that they use 1% of training data to train a Ranking SVM to generate initial ranked list of items, upon which clicks are simulated using different click models. How does the usage of more/less portion of training data affect the comparative baselines and in turn the relative performance improvement provided by the proposed approach over those baselines?



**Questions:**

- Could the authors address the questions raised in the weaknesses section of the review?

**Limitations:**

- Authors have not explicitly addressed limitations of their approach or any potential negative societal impact in the manuscript. However, authors' proposed approach is not a significant departure from the prevalent large scale recommender systems.

---

> ### Author Rebuttal · Authors · 2023-08-10
>
> We thank the reviewer for the appreciating our motivation, presentation and experiments.
>
> **Q:** Is this a Markov Decision Process? Given the authors' approach uses state representation, it is unclear to me how/if the Markovian property Pr(st|s0,a0 …st-1, at-1) = Pr(st| st-1, at-1) still holds?
>
> **A:** We would like to clarify that according to the state definition in Eq (3), the Markovian property holds. Specifically, the state only depends on the current position and previous documents.
>
> **Q:** Effect of Click Data Generation. How does the usage of more/less portion of training data affect the comparative baselines and in turn the relative performance improvement provided by the proposed approach over those baselines?
>
> **A:** Thank you for the question. Please refer to Table 1 and our answer in the general response to all reviewers for new experimental results. Due to time limit, we have compared CUORL-CQL against different logging policies and will add more baselines to the experiment in next version.

---

> > ### Comment · Reviewer_fhCJ · 2023-08-14
> >
> > Dear authors,
> >
> > Thank you for your response and additional experimental results presented in the rebuttal document. I shall take this additional information into account during the discussion phase.
> >
> > Thank you.

---

> > > ### Author Response · Authors · 2023-08-21
> > > **Thank you**
> > >
> > > We sincerely thank the reviewer for the time and effort in reviewing our paper.
> > >
> > > Regards,
> > >
> > > Authors

---

### Official Review · Reviewer_XmTH · 2023-07-23

**Soundness:** 3 good
**Presentation:** 2 fair
**Contribution:** 3 good
**Rating:** 5
**Confidence:** 3

**Summary:**

In this work, the authors model the process of learning optimal ordering of ranked lists, called Learning to Rank (LTR) as a Markov decision process (MDP). This allows them to utilize techniques from reinforcement learning to solve for the policy that generates the optimal ordering. In practical applications of LTR, a pre-collected dataset (offline) is typically used which is collected using a logging policy that is not optimal (off-policy). As a result, techniques for offline off-policy reinforcement learning are used in this work.
In the MDP formulation, the reward is defined as an item of the ranked list being clicked. The distribution of clicks is referred to as a click model, which results in various reward functions. An RL algorithm can optimize a wide range of reward functions and allows this approach to be largely click-model agnostic.
The approach is empirically validated on semi-synthetically generated click datasets, and it demonstrates competitive performance across different click models compared to click-model-specific baselines for learning optimal rankings.


**Strengths:**

The paper succinctly models LTR as a sequential decision-making problem, building on top of prior work,  in a manner that allows the use of transformer architecture to be employed in conjunction with an RL algorithm. This allows for a couple of benefits: 1) being agnostic of specific click models since any click distribution in the offline dataset can in theory be encoded as the reward that the RL algorithm can optimize for, and 2) allowing for flexibility in defining the state representation through the use of attention and position embeddings.
Both of the claims are supported by strong empirical analysis.


**Weaknesses:**

The paper extended prior formulations of LTR as an MDP to allow for using the transformers architecture, with the aim performing policy learning without click model specific methods. In addition to the existing experiments, one experiment that would highlight the point would be one that learns the optimal ranking for data generated from a non-standard click model – say a randomly picked click distribution – where all other methods fail.

Additionally, the utility of the section on the optimality of rankings (Definition 3; Assumption 3.1) is unclear. These are possibly included to tie this work to existing literature, however, this method should not require those conditions for finding an optimal policy.
On a related note, L122-L123 incorrectly states that Definition 1 is explained in Appendix C, which only discusses the definitions of examination probabilities of various click models.

The writing and presentation could be improved in places, a few examples being highlighted below.
Minor/Typos:
- “Debiasing” as caused by bias in the data, versus bias due to estimation procedure has been used interchangeably through the paper leading to a fair amount of confusion (Ex: L26-L29)
- Equation (1) should have a $\propto$ and not $=$
- L121: mutually independent of what?
- L227: $d\\_model$ undefined.
- Algorithm 1: Mistake on Line 7


**Questions:**

A few questions:
- Assumption 3.1 holds only for monotonically decreasing examination probabilities. The reference [64] proves the result in Assumption 3.1 only for PBM and cascade models, and is not the case for other click models. Why is this top-down scanning of the ranked list necessary for the working of the algorithm?
- The robustness to change in $\alpha$ in CQL seems to indicate that just fitting a Q function on the offline data should suffice. This would be true if the logging policy provides enough support for all actions, which is commonly the case in practice where uniform random logging policies are used. Why is a conservative algorithm (CQL) necessary for the experiments?
- (L148-L149) How is the action set restricted to not repeat actions implementationally?
- The gains in performance are more significant for ERR as compared to NDCG. Why is that the case?


**Limitations:**

Like any other method for LTR, the performance of the algorithm should depend on the coverage provided by the logging policy. A comment on the effect of the coverage of the logging policy and it's effects on the learnt rankings would be useful.

State representations play an important role in the performance of the algorithm and in the context of this work the representations are learned via architectural choices in the Q-function estimator. The claim that (L248) that an additional component is added on top of CQL is overstated.

---

> ### Author Rebuttal · Authors · 2023-08-10
>
> We thank the reviewer for appreciating the advantages of our click model-agnostic solution.
>
> **Q:** The paper extended prior formulations of LTR as an MDP to allow for using the transformers architecture, with the aim of performing policy learning without click model specific methods. In addition to the existing experiments, one experiment that would highlight the point would be one that learns the optimal ranking for data generated from a non-standard click model – say a randomly picked click distribution – where all other methods fail.
>
> **A:** Thank you for the suggestion of testing on non-standard click models such as randomly picked click distribution. According to Definition 1, the click model at any position should only be determined by the rank of the current position and previous documents, where a randomly picked click distribution may not satisfy this assumption. We do want to point out this assumption is not restrictive as the classic PBM and CASCADE model all satisfy this, and we have already evaluated on several more complicated click models including UBM, DCM, and CCM and reported the results in the main paper and Appendix.
>
> **Q:** Additionally, the utility of the section on the optimality of rankings (Definition 3; Assumption 3.1) is unclear. These are possibly included to tie this work to existing literature, however, this method should not require those conditions for finding an optimal policy. On a related note, L122-L123 incorrectly states that Definition 1 is explained in Appendix C, which only discusses the definitions of examination probabilities of various click models.
>
> **A:** Definition 3 is used to define the optimality of rankings. Note that the goal of LTR is to find the optimal ranking list while the objective of RL method is the optimal value function. We want to emphasize that these two optimality are not necessarily aligned, and they are only equivalent when Assumption 3.1 is satisfied. For most classical click models, such as PBM and cascade model Assumption 3.1 is satisfied and we can show the equivalence between the two. Sorry for the confusion in L122-L123. As we can see in Appendix C, we explained the specific form of various examination probabilities, while the attractivenesses $\alpha(\cdot)$ only depend on the documents and are the same across all the click models. By showing these forms, we are explicitly showing how these models are instances of Definition 1 and could be incorporated into our unified framework.
>
> **Q:** Assumption 3.1 holds only for monotonically decreasing examination probabilities. The reference [64] proves the result in Assumption 3.1 only for PBM and cascade models, and is not the case for other click models. Why is this top-down scanning of the ranked list necessary for the working of the algorithm?
>
> **A:** As we have explained above, the optimality assumption in Assumption 3.1 is needed, as the optimal value in RL may not give an optimal ranking list. It is unclear if click models with non-monotonically decreasing examination probabilities can be solved by our framework since optimizing rewards may not align with optimizing ranking list. We empirically showed that besides PBM and cascade model, major click models like UBM, DCM, CCM (all have monotonically decreasing examination probabilities) can be effectively solved by our offline RL framework. It would be interesting to look beyond this assumption in future.
>
> **Q:** (L148-L149) How is the action set restricted to not repeat actions implementationally?
>
> **A:** Each action is a query-document pair, which is unique in the dataset.
>
> **Q:** The gains in performance are more significant for ERR as compared to NDCG. Why is that the case?
>
> **A:** Compared to NDCG, ERR is more sensitive to the relevance of top results. This observation suggests our improvements are more beneficial to the top positions of ranking list compared to baselines.

---

> > ### Comment · Reviewer_XmTH · 2023-08-13
> >
> > The authors' response addresses some of my concerns while overlooking a couple of others.
> >
> > My question about the non-standard click model still stands: The method is designed to be click-model agnostic, and should in theory be able to optimize for any reward --- click probability --- independent of Definition 3. In that setting, all other click-model specific baseline would be expected to struggle. This would be an insightful result about the utility of the method.
> >
> > I will keep my initial score.

---

> > > ### Author Response · Authors · 2023-08-14
> > >
> > > We thank the reviewer for the follow-up comment and are glad to hear that some concerns have been addressed by our initial response.
> > >
> > > The reviewer's comment on
> > > > The method is designed to be click-model agnostic, and should in theory be able to optimize for any reward --- click probability --- independent of Definition 3.
> > >
> > > caught our attention. Firstly, we would like to emphasize that we have accurately pointed out that our method can be applied to "a wide range of click models" as early as in the abstract. It would be a misunderstanding and an overstatement that our method (or any algorithm) can solve offline LTR with click feedback without basic assumptions of how clicks are generated.
> > >
> > > Secondly, our Definition 3 and Assumption 3.1 are mild but necessary assumptions on the optimality of the ranking list. Definition 3 assumes optimal ranking is the one with descending attractiveness of documents, which is the standard assumption of LTR problem and aligns with evaluation metrics such as ERR and NDCG. Assumption 3.1 assumes optimal ranking list maximizes total rewards (clicks), which is needed to align the goal in Definition 3 and the solution of RL. The assumption has been proved correct for PBM and Cascade model [1] and we empirically showed that our method worked well for other click models such as DCM, CCM and UBM. However, for certain click models, optimal ranking list does not maximize total clicks. We here provide a counterexample: consider three documents with attractiveness $d_1 = 0.1, d_2=0.2, d_3=0.3$ and the examination probabilities depend on both document and position. For the two ranking list
> > > * attractiveness: 0.3 0.2 0.1, examination prob: 0.5, 0.1, 0.1
> > > * attractiveness: 0.1 0.2 0.3, examination prob: 0.5, 0.4, 0.3
> > >
> > > , the optimal ranking list (first one) has smaller expected total clicks (0.18 < 0.22) because this click model encourages postponing good documents to keep user browsing and generating more clicks. Thus this example cannot be solved by our method. However, we note that this example does not belong to any click models we studied (PBM, Cascade, DCM, CCM, UBM) and to the best of our knowledge, has not been studied in previous offline/counterfactual LTR literature. Actually, our method can already cover most of the click models that have been studied in offline LTR.
> > >
> > > We are happy to answer any questions that the reviewer finds not fully addressed.
> > >
> > > [1] Masrour Zoghi, Tomas Tunys, Mohammad Ghavamzadeh, Branislav Kveton, Csaba Szepesvari, and Zheng Wen. Online learning to rank in stochastic click models. In International Conference on Machine Learning, pages 4199–4208, 2017.

---

> > > > ### Comment · Reviewer_XmTH · 2023-08-14
> > > >
> > > > Thank you for the clarification. This along with the additional experiments address my concerns, and accordingly, I have updated my score.

---

> > > > > ### Author Response · Authors · 2023-08-21
> > > > > **Thank you for the discussion**
> > > > >
> > > > > We are happy to see that our response addressed the concerns. We thank the reviewer for the review and engagement during the discussion period.
> > > > >
> > > > > Regards,
> > > > >
> > > > > Authors

---

### Author Rebuttal · Authors · 2023-08-10

We thank all the reviewers for your detailed comments and questions, which helped us to improve our paper. We appreciate that the reviewers generally agree our unified formulation is well-motivated, our click model-agnostic approach is novel and original, and experimental results are extensive and support the claim. We have responded to your individual comments and will add the following new results and discussion in the paper.

**CQ1:**   Performance Comparison of CQL and SAC (by Reviewer XmTH, 79J3, ynTd).

**A1:** We first emphasize that the main contribution of the paper is formulating off-policy learning to rank (LTR) problem as an MDP and proposed an offline RL framework that leverages **any** off-the-shelf RL algorithm as the plug-in solver. We choose the popular CQL algorithm to conduct experiments as it can alleviate over-estimation caused by distributional shift with conservative Q-learning. However, as reviewers have noticed from the results in main paper, SAC (which is equivalent to CQL with conservative parameter $\alpha=0$) performs closely to CQL with the initially chosen fixed $\alpha=0.1$.  To further investigate this, we report the performance of SAC and CQL with optimal $\alpha$ in Table 2 of the PDF, where we search for optimal value separately for each click model. We observe that the performance of CQL with optimal $\alpha$ is much higher than simple SAC especially in NDCG. This observation suggests that distribution shift and over-estimation are still an issue in the off-policy LTR setting, and making CQL work needs additional effort of careful tuning, which is the case for other domains as well. In this paper, we have shown that the off-policy LTR problem can be unified and solved by our click-model agnostic offline RL framework with simple RL solvers like CQL and SAC. Comparing the pros and cons of various advanced RL algorithms in off-policy LTR is beyond the scope of this paper, but an interesting follow-up.

**CQ2:** The impact of logging policy (by Reviewer XmTH, fhCJ)

**A2:** We conduct new experiments on the impact of logging policies on Web10k dataset. On each click model, we compare three different logging policies: SVMRank trained with 1% training data, SVMRank trained with 0.01% data, and random policy. The result of CUOLR-CQL with fixed $\alpha=0.1$ is reported in Table 1 of the PDF.  Not surprisingly, we can see that with a worse logging policy, the performance of learned policy also decreases. This also highlights the need for a good logging policy in offline LTR.

---

> ### Author Response · Authors · 2023-08-15
> **More experimental result**
>
> To verify the robustness of our proposed method CUOLR-CQL, we add the performance comparison of CUOLR-CQL with various baselines under logging policies with different qualities. Specifically, as stated in A2, we add a logging policy with SVMRank trained with 0.01% data. The performance when using a logging policy with SVMRank trained with 1% training data is shown in the main paper Table 1. The performance are shown in the following table:
>
> The metrics from left to right are: ERR@3,5,10, NDCG@3,5,10
>
> DLA (PBM): | 0.155 | 0.179 | 0.202 | 0.246 | 0.261 | 0.293 |
>
> IPW (PBM): | 0.164 | 0.187 | 0.210 | 0.259 | 0.273 | 0.304 |
>
> Ours(PBM): | 0.214 | 0.239 | 0.262 | 0.324 | 0.339 | 0.367 |
>
> DLA (CASCADE): | 0.171 | 0.195 | 0.218 | 0.261 | 0.277 | 0.309 |
>
> IPW (CASCADE): | 0.180 | 0.204 | 0.227 | 0.284 | 0.297 | 0.327 |
>
> Ours(CASCADE): | 0.214 | 0.239 | 0.262 | 0.326 | 0.340 | 0.368 |
>
> DLA (DCM):  | 0.191 | 0.215 | 0.238 | 0.290 | 0.304 | 0.333 |
>
> IPW (DCM):  | 0.190 | 0.214 | 0.237 | 0.299 | 0.310 | 0.340 |
>
> Ours (DCM): | 0.212 | 0.238 | 0.260 | 0.324 | 0.338 | 0.367 |
>
>
> This demonstrates the effectiveness and robustness of our proposed method under various logging policies. We thank the reviewer for the suggestion here and will also add this table in the final version of the paper.

---

### Decision · Program_Chairs · 2023-09-21

**Decision:**

Accept (poster)

**Comment:**

Reviewers appreciate the overall perspective of learning to rank (LTR) as an offline RL problem learned from logged click data.  While the idea of viewing LTR as an MDP is not new, the authors claim that learning from implicit feedback (click data) and the ability to leverage various click models under a unified framework is new.  Reviewers generally agree with these claims of novelty and believe this approach and use of offline RL may inspire follow-on research.  The rebuttal discussion and additional experimental results had a positive impact on reviewer opinion and there was a general consensus among all five reviewers that the paper could be accepted, though some reviewers maintain residual concerns post-rebuttal that should be addressed in the final revision.

In general, the authors are requested to incorporate points of clarification from the post-rebuttal discussion.  In addition, reviewers expressed specific concerns during final decision discussion that the following key points need to be included on revision:
- Overstatement of claims (about state representations and work in the Appendix) need to be corrected.
- Additional experiments and comments presented in the rebuttal (CQL, effect of the logging policy) need to be included (in the main paper or Appendix) and discussed in the main paper.
- Clarification of the Markovian property -- I include the reviewer's comment verbatim: "Authors representation of the state takes into account the sequence of documents, as against just the set of documents viewed by the user because the authors concatenate the representations of prior states (an order sensitive aggregation) as against a summation or mean representation of prior states (an order insensitive aggregation). In my opinion, that invalidates authors' claim about this setup being an MDP because the next state of the agregated MDP depends on the sequence of prior states, not just an aggregation of the prior states. This means that Markovian property does not hold, at least when it comes to vanilla MDP. The authors could possibly prove that it is the case for a higher order Markov Chain, but they have not."